



# Assessment of Irrigation Physics in a Land Surface Modeling Framework using Non-Traditional and Human-Practice Datasets

Patricia M. Lawston[1,2], Joseph A. Santanello, Jr. [2], Trenton E. Franz[3], Matthew Rodell [2]

[1]Earth System Science Interdisciplinary Center, University of Maryland, College Park, MD 20740, USA

[2]Hydrological Sciences Laboratory, NASA Goddard Space Flight Center, Greenbelt, MD 20771, USA
[3]School of Natural Resources, University of Nebraska-Lincoln, Lincoln, NE 68583, USA

*Correspondence to*: Patricia M. Lawston (patricia.m.lawston@nasa.gov)

**Abstract.** Irrigation increases soil moisture, which in turn controls water and energy fluxes from the land surface to the

planetary boundary layer and determines plant stress and productivity. Therefore, developing a realistic representation of

irrigation is critical to understanding land-atmosphere interactions in agricultural areas. Irrigation parameterizations are

becoming more common in land surface models and are growing in sophistication, but there is difficulty in assessing the

realism of these schemes, due to limited observations (e.g., soil moisture, evapotranspiration) and scant reporting of

irrigation timing and quantity. This study uses the Noah land surface model run at high resolution within NASA's Land

Information System to assess the physics of a sprinkler irrigation simulation scheme and model sensitivity to choice of

irrigation intensity and greenness fraction datasets over a small, high resolution domain in Nebraska. Differences between

experiments are small at the interannual scale, but become more apparent at seasonal and daily time scales. In addition, this

study uses point and gridded soil moisture observations from fixed and roving Cosmic Ray Neutron Probes and co-located

human practice data to evaluate the realism of irrigation amounts and soil moisture impacts simulated by the model. Results

show that field-scale heterogeneity resulting from the individual actions of farmers is not captured by the model and the

amount of irrigation applied by the model exceeds that applied at the two irrigated fields. However, the seasonal timing of

irrigation and soil moisture contrasts between irrigated and non-irrigated areas are simulated well by the model. Overall, the

results underscore the necessity of both high-quality meteorological forcing data and proper representation of irrigation for

accurate simulation of water and energy states and fluxes over cropland.



## 1 Introduction

Irrigation is vital to feeding the world's population, accounting for ~40% of global food production and 20% of arable land (Molden 2007; Schultz et al., 2005). Approximately 70% of global freshwater withdrawals (FAO, 2010) are used to meet the demand for irrigation, thereby altering the hydrologic cycle and raising questions about water resources sustainability. As a

result, irrigation modeling studies have sought to understand the impacts of irrigation on ambient weather (Sorooshian et al., 2011, 2012), precipitation and streamflow (Harding and Snyder 2012a,b; Kustu et al., 2011), and regional to global climate (Lo and Famiglietti, 2013; Puma and Cook, 2010). Although the atmospheric response is often sensitive to the details of the irrigation scheme used in modeling studies, the observational data needed to fully vet an irrigation scheme (e.g., irrigation timing, practices, and co-located soil moisture) are generally not obtainable at the scale of LSMs, making robust evaluation

difficult and casting doubt on conclusions about downstream impacts on regional weather, precipitation, and long term climate.

The impact of water resources management practices such as irrigation on the water cycle is significant enough that the World Climate Research Program (WCRP) has identified anthropogenic changes to the continental water cycle as a Grand Science Challenge to be addressed over the next 5 to 10 years (Trenberth and Asrar, 2014). In response, the Global Energy

and Water Cycle Exchanges project's (GEWEX) Hydroclimatology Panel (GHP) and Global Land/Atmosphere System Study (GLASS) have begun a joint effort to advance the representation of human water resources management in land surface and coupled models (van Oevelen, 2016). To effectively meet these challenges, new, non-traditional datasets are needed to evaluate and improve representation of irrigation in models and to assess the processes by which simulated irrigation impacts the water cycle.

The work presented here touches on each of these issues by comprehensively assessing a sprinkler irrigation algorithm in a land surface model (LSM) and evaluating the results with both conventional and non-traditional datasets. The paper is organized in the following way: Sect. 2 provides relevant background on recent irrigation modeling efforts with an emphasis on differences in irrigation schemes and previous evaluation efforts, and introduces gridded soil moisture from the Cosmic-Ray Neutron Probe method (CRNP) as a potential tool for evaluation of land surface model irrigation. A description of the

experimental design, including the land surface modeling framework and the irrigation algorithm, are presented in Sect. 3.





Sect. 4 describes the results, first in the context of model sensitivity and secondly through an evaluation of the model simulations with observations. A discussion of the results and the applicability of this study to future irrigation modeling efforts are discussed in Sect. 5, and conclusions are stated in Sect. 6.

## 2 Background

### 2.1 Irrigation physics

Irrigation increases soil moisture and therefore has the potential to influence local and regional clouds, precipitation, and ambient weather via land-planetary boundary layer (PBL) coupling processes (Santanello et al., 2011). By increasing latent and decreasing sensible heat fluxes, near surface temperature is reduced within irrigated areas (Bonfils and Lobell, 2007; Kanamaru and Kanamitsu, 2008). The irrigation-modified land energy balance alters the proportion of heat and moisture

contributed to the PBL, thereby influencing PBL growth and entrainment (Kueppers and Snyder, 2011; Lawston et al., 2015). As a result, the PBL over irrigated areas is often shallower and moister, potentially resulting in alterations to convective cloud development (Adegoke et al., 2007; Qian et al., 2013). Irrigation applied over large areas not only affects local ambient weather, but models indicate that it can also modify precipitation patterns in areas remote from the source (de Vrese et al. 2016), which can further alter streamflow (Kustu et al. 2011). Extensive irrigation projects, such as the Gezira

Scheme in East Africa, have been shown to influence regional weather by changing circulation and precipitation patterns (Alter et al., 2015).

These significant potential impacts of irrigation on temperature, clouds, precipitation, and related fluxes necessitate an appropriate representation of irrigation in coupled land-atmosphere models. This need has been addressed via irrigation parameterizations in LSMs that largely fall into three types of schemes: 1) defined increases to soil moisture in one or more

soil layers (Kueppers and Snyder, 2011; de Vrese et al. 2016), 2) the addition of water as pseudo-precipitation to mimic sprinkler systems (Ozdogan et al., 2010; Yilmaz et al., 2014), and 3) modifications to vapor fluxes as a proxy for increased evapotranspiration resulting from highly efficient (e.g., drip) irrigation (Douglas et al., 2006; Evans and Zaitchik, 2008). These schemes are generally dependent on parameter input datasets and user defined thresholds, affording a degree of customization, but also introducing uncertainty and potential error. Model sensitivity to the selection of datasets and

thresholds is not trivial, as differences can alter the magnitude of irrigation-induced changes to the water and energy budgets.



For example, a flood irrigation parameterization with a two different triggering thresholds resulted in up to 80 W m$^{-2}$ difference in average seasonal latent heat flux increase in the U.S. Central Great Plains (Lawston et al., 2015). In another case, Vahmani and Hogue (2014) tested several irrigation demand factors and irrigation timing in their urban irrigation module, finding fluxes, runoff, and irrigation water are sensitive to both inputs. Additionally, the same parameterization used

in a different model (Kueppers et al., 2008; Tuinenburg et al., 2014), or in the same model but at a different resolution (Sorooshian et al., 2011) has also produced different coupled atmospheric impacts.

## 2.2 Evaluation of irrigation in LSMs

The sensitivity of atmospheric predictions to the details of the irrigation scheme makes it imperative to systematically evaluate irrigation parameterization, datasets, and thresholds in a controlled modeling study to determine the levels of

uncertainty in the perturbation and subsequent results. However, datasets required for evaluation, such as irrigation amount, irrigation timing, and co-located continuous soil moisture observations, are not widely available, making it difficult to evaluate irrigation schemes (Kueppers et al., 2007). Modeling studies that have included some assessment of the irrigation scheme have used comparisons to annual water withdrawals for irrigation (Lobell et al., 2009; Pokhrel et al., 2012), outdoor water use (Vahmani and Hogue, 2014), recommended amounts of irrigation (Sorooshian et al., 2011, 2012), or irrigation

water usage reported by the U.S. Geological Survey (Ozdogan et al., 2010). Bulk estimates, such as these, are often not used for robust evaluation, but rather indicate that the simulated results are reasonable.

In some cases, additional analysis of the observations has been successful in converting estimates to quantities usable for comparison. For example, Pei et al. (2016) used a potential evapotranspiration ratio to estimate June, July, and August irrigation usage from USGS yearly county-level estimates in order to validate irrigation amounts in the WRF-Noah Mosaic

coupled model. The study found good agreement between the amounts simulated and that of the modified observations at 30 km horizontal resolution. In other cases, county and coarser resolution irrigation estimates have been used to constrain the irrigation algorithm output. Leng et al. (2013, 2014) calibrated the irrigation scheme in the Community Land Model (CLM) to reproduce county and water resources region irrigation water usage, respectively. Taken together, these studies exhibit recent progress made in irrigation modeling evaluation at regional to continental scales, but the datasets employed are

insufficient for evaluation at high resolution and shorter (e.g. season to sub-monthly) time-scales.



As soil moisture is the primary control over fluxes and vegetation health, an evaluation of soil moisture sensitivity to irrigation is equally as important as realistic irrigation estimates. Such evaluation is challenging as it demands soil moisture observations that are temporally and spatially continuous and at high enough resolution to resolve an irrigation signal. Satellite remote sensing has obvious potential to reach these goals, but retrievals of soil moisture are generally too coarse

(i.e., ~25-40 km spatial resolution) and exhibit limited skill, at best, in detecting an irrigation signal (Kumar et al., 2015). At the other spatial extreme, point observations of soil moisture values are not representative of the larger area average (Entin et al., 2000). The aggregation of these observations into homogeneous, quality controlled datasets, such as the North American Soil Moisture Database (NASMD, Quiring et al. 2016) and the International Soil Moisture Network (ISMN, www.ipf.tuwien.ac.at/insitu), are promising for LSM evaluation more broadly, but in-situ measurements in irrigated fields,

needed for irrigation scheme evaluation, are still sparse.

**2.3 Cosmic-ray neutron probe (CRNP)**

A potential solution to fill the gap between point and remote sensing observations of soil moisture is the Cosmic-Ray Neutron Probe (CRNP) method, organized through the Cosmic Ray Soil Moisture Observing System (COSMOS, Zreda et al., 2012), which has ~200 probes operating globally since 2011. CRNP is a new and novel way to obtain high-resolution,

semi-continuous soil moisture observations, and as a result, has the potential to advance LSM and irrigation parameterization development. The CRNP is placed above the ground and measures neutrons produced by cosmic rays in the air and soil over a diameter of 300+/- 150 m, depending on atmospheric pressure and humidity (c.f. Desilets and Zreda, 2013 and Kohli et al., 2015). The neutron density measured by the probe is inversely correlated with soil moisture and can be calibrated using local soil samples to an error of less than 0.03 $m^3$ $m^{-3}$ (Franz et al., 2012). The result is reliable, area-average soil water content

integrated to a depth of ~20-40 cm, depending on water content, bulk density, and lattice water, available at the same spatial scale as LSMs (Franz et al., 2012) .

The characteristics of the CRNP, including the non-contact, passive data collection, make the CRNP portable and able to collect data while in motion. Desilets et al. (2010) first used a roving CRNP in Hawaii to obtain transects of soil moisture at highway speeds. More recently, Chrisman and Zreda (2013) and Dong et al. (2014) used the roving CRNP at the mesoscale

in Arizona and Oklahoma. Franz et al., (2015) mounted a large CRNP instrument to the bed of a pickup truck and completed



roving surveys during the growing season of 2014 in a 12 x 12 km area of eastern Nebraska. The instrument collected ~300

neutron counts every minute and was driven at a maximum speed of 50 km per hour, allowing for high resolution maps to be

generated via geostatistical interpolation techniques. The spatial locations of each neutron measurement are given by the

midpoint of successive rover locations and together are spatially interpolated via kriging to 250 m resolution. The surveys

were completed every 3-4 days from May to September. In addition, 3 fixed probes were located inside the domain

continuously recording soil moisture. Franz et al. (2015) used the fixed and roving data with a simple merging technique to

produce 8-hour soil moisture products at 1, 3, and 12 km resolutions.

The work presented here uses the data and products gathered and generated in Franz et al. (2015) for evaluation of a

sprinkler irrigation algorithm in a LSM environment, described in Sect 3. Specifically, the data are available for the 2014

growing season and include: timing and amount of irrigation water applied at two sites (one maize, one soybean), soil water

content from a stationary COSMOS probe at these two irrigated sites, plus a rainfed site of mixed soybean and maize, and

lastly, high-resolution gridded soil moisture at 3-4 day temporal resolution during the growing season (May to Sept) from the

CRNP rover. The integration of human practice data (irrigation amount), physical observations (soil moisture point and

spatial observations), and model simulations to evaluate the sprinkler algorithm and its impacts on soil moisture is a key and

novel feature of this study. The main goals of this work are first to assess the physics of the simulated sprinkler irrigation,

and secondly to evaluate the realism of the irrigation amounts and impacts to soil moisture.

### 3. Methods

### 3.1 Models and experimental design

NASA's Land Information System (LIS; Kumar et al., 2006) is used in this study to assess the performance of the Sprinkler

irrigation scheme. LIS is a land surface modeling and data assimilation system that allows users to choose from a suite of

land surface models which can then be run offline while constrained and forced by best available surface and satellite

observations. LIS can be fully coupled to the Weather Research and Forecasting model (WRF, Skamarock et al. 2005) in the

NASA Unified WRF (NU-WRF, Peters-Lidard et al. 2015) framework. This configuration, LIS-WRF, has been used at the




regional scale to assess the downstream impacts of irrigation on the PBL, but the performance of the irrigation scheme was not assessed (Lawston et al. 2015).

In this study, the Noah land surface model (Chen et al., 2007) version 3.3, was run offline within the LIS framework at 1 km spatial resolution over a 15 x 15 km area in eastern Nebraska, near the town of Waco. The size and location of the domain

were designed to encompass the study area of Franz et al. (2015) to make use of the CRNP rover data, human practice information, and point and spatial observations produced by their work, as discussed in Sect. 2.

The LIS simulations were run for 6 years (1 Jan 2009 to 31 Dec 2014) yielding daily output. The long-term simulation output was used to initialize restart-simulations for the growing seasons of 2012 and 2014 to produce hourly output for more detailed investigation during these periods, and the 3-5 year spinup periods, respectively, were shown to be sufficient for this

region (Lawston et al. 2015). The analysis focuses on these two years (i.e., 2012 and 2014) to evaluate the irrigation algorithm during contrasting antecedent soil moisture conditions (e.g., relatively dry and wet, respectively), and to assess the performance of the scheme using the CRNP observations available in 2014.

To capitalize on the controlled nature of the study area and the irrigation scheme's dependence on green vegetation fraction (GVF) and irrigation intensity, discussed in detail in section 3.2, four types of simulations were completed and will hereafter

be referred to as the 1) Control, 2) Standard, 3) Tuned, and 4) SPoRT simulations. The Control run is the only simulation that has the irrigation scheme turned off. The Standard simulation differs from Control only in that the sprinkler irrigation scheme is turned on and the Global Rainfed, Irrigated, and Paddy Croplands (GRIPC; Salmon et al., 2015) dataset is used to prescribe irrigation intensity at 1km resolution needed for the sprinkler algorithm. The GRIPC dataset irrigation intensity is unrealistically high in the study area, as evidenced by only 5% of the gridcells having intensity less than 100% (Fig. 1). To

correct for this overestimation, the Tuned simulation uses an irrigation intensity map created by reducing the GRIPC irrigation intensity according to a land use map generated from ground truth observations (Franz et al. 2015), thereby more accurately reflecting irrigation patterns in the study area (i.e. observationally tuned; Fig. 1). The SPoRT run makes use of the GRIPC irrigation intensity dataset, like the Standard run, but uses a real-time GVF product from NASA-Marshall's Short Term Prediction, Research, and Transition Center (SPoRT; Case et al., 2014). This is in contrast to the other runs that use

climatological GVF from the National Centers for Environmental Prediction (NCEP).

The SPoRT GVF is created using normalized difference vegetation index (NDVI) from the Moderate Resolution Imaging

Spectroradiometer (MODIS) onboard the Terra and Aqua satellites and as such reflects the vegetation response to

temperature and precipitation. In this way, the SPoRT GVF dataset captures interannual variability in vegetation that is

missed by climatological GVF (Figure 2). Additionally, SPoRT GVF has higher spatial resolution (i.e., 3 km vs. ~16km for

climatology) and has been shown to improve the simulated evolution of precipitation in a severe weather event as compared

to GVF from climatology when using LIS coupled to a numerical weather prediction model (Case et al., 2014). The use of

the SPoRT GVF dataset can be viewed as a middle-of-the-road approach between a simple representation of vegetation (e.g.,

climatology) and more sophisticated, but computationally-expensive methods, such a dynamic vegetation or crop growth

models (e.g. Harding et al., 2015; Lu et al., 2015). As the SPoRT dataset is not available prior to 2010, the long-term SPoRT

simulation uses climatological GVF for 2009-2010, and the SPoRT GVF dataset is incorporated in December 2010 and used

throughout the rest of the simulation.

Additional datasets common to all simulations include MODIS – International Geosphere Biosphere Program (MODIS-

IGBP) land cover, State Soil Geographic (STATS-GO) soil texture, University of Maryland (UMD) crop type, and National

Land Data Assimilation System – Phase 2 (NLDAS2; Xia et al., 2012) meteorological forcing that includes bias corrected

radiation and gauge-based precipitation.

### 3.2 Irrigation scheme

The preferred method of irrigation in Nebraska is the center pivot sprinkler system (NASS, 2014), and as such, we evaluate

the sprinkler irrigation algorithm in LIS. The sprinkler scheme is described in Ozdogan et al. (2010) and was preliminarily

tested and compared against two other irrigation schemes (drip and flood) available in LIS in Lawston et al. (2015).

Sprinkler applies irrigation as precipitation when the root zone moisture availability falls below a user-defined threshold. In

this study, we use a threshold of 50% of the field capacity, after Ozdogan et al. (2010).

In an effort to reproduce appropriate timing and placement of irrigation, a series of model checkpoints must be passed to

allow for irrigation triggering. These checkpoints essentially boil down to four main questions:

1) Is the land cover irrigable?

2) Is there at least some irrigated land?



3) Is it the growing season?

4) Is the soil in the root zone dry enough to require irrigation?

The first two questions invoke direct tests against the static datasets (land cover and irrigation intensity, respectively), while the remaining two questions require additional calculations involving one or more time-varying datasets. The growing

season, addressed in question three, is a function of the gridcell GVF as described in Ozdogan et al. (2010) and results in a season that spans roughly June through September in the study area. The last question, the determination of irrigation requirement, is dependent on two main features – the soil moisture and the definition of the root zone. Soil moisture is influenced by the meteorological forcing (e.g., how much rain falls and where) and soil texture (e.g., how long the moisture sticks around), while the root zone is the product of the maximum root depth (as defined by crop type) scaled by the GVF to

mimic a seasonal cycle of root growth. Taken together, this means that the irrigation scheme is primarily controlled by six datasets: landcover, irrigation intensity, soil texture, crop type, meteorological forcing, and GVF.

For this limited study area, the land cover, crop type, and soil texture are homogenous throughout the domain as given by the input datasets (croplands, maize, and silt loam, respectively), meaning any heterogeneity in irrigation amounts and impacts can be attributed to only the meteorological forcing, GVF, and irrigation intensity. As the meteorological forcing is the same

for all simulations, the experimental design leverages the unique characteristics of the controlled domain to assess the sensitivity of the irrigation algorithm specifically to changes in irrigation intensity and GVF; two important and common datasets in irrigation modeling. The irrigation algorithm is assessed first in the context of its physical response to forcing at the interannual, seasonal, and daily scales, and secondly, the results are evaluated against available observations in the growing season of 2014 (i.e., model performance).

**4. Results**

**4.1 Model sensitivity at the interannual scale**

Figure 3 shows the domain and monthly averaged irrigation amount applied for each of the three irrigation runs over the full six-year period. Interannual variability in the background precipitation (i.e., summer drought or pluvial periods) is reflected in the irrigation requirement, with dry seasons, such as 2012, exhibiting large irrigation demand, while wet seasons like 2011

and 2014 result in markedly less water applied. The average irrigation amount varies little between the experiments at this scale, around 1 mm day$^{-1}$, but a few features of the dataset differences are apparent. The irrigation algorithm scales the amount of water applied by multiplying with the irrigation fraction value. The GRIPC irrigation dataset has greater irrigation intensity values everywhere in the domain, and as a result, the Standard run always applies more water than Tuned. The

SPoRT run is less consistent in relation to the other methods; at times applying more water than both methods (e.g. July 2012), at others applying less (e.g. Sep 2012). This behavior is determined by the relative magnitude of the SPoRT GVF as compared to climatological GVF (Figure 2), as the GVF scales the root zone such that more water is applied by the irrigation scheme to more mature crops.

Figure 4 shows the percent change from control in soil moisture for each of the irrigation runs and each model soil layer.

Irrigation increases soil moisture in all soil layers and all simulations. Increases in the third soil layer are quite consistent annually with a near doubling of the soil moisture when irrigation is turned on. The top and second layer fluctuations resemble the irrigation amount time series, indicating that the top two layers are more sensitive to the amount of irrigation water applied. These layers respond more quickly to irrigation, while percolation, and therefore time, is needed to impact the deeper soil layers. Differences between the irrigation runs are virtually undetectable in the top and second layers, but the

cumulative impact of the differences in irrigation amounts and timing are reflected in differences in the third soil layer. The third and fourth layers are deeper and thicker (0.6 m and 1.0 m thickness, respectively) and as such are able to hold more water than the top and second layers (0.1 and 0.3 m thickness).

### 4.2 Model sensitivity at the seasonal scale

Figure 5 shows the average daily change from control in latent (Qle) and sensible (Qh) heat fluxes (left axis) as well as the

daily precipitation amount from the NLDAS-2 meteorological forcing data (right axis) for May-October 2012 and 2014. Limited rainfall throughout the 2012 season resulted in the triggering of irrigation frequently throughout the growing season, including a stretch through July and August where irrigation was triggered somewhere in the domain every day (not shown). The 2014 growing season featured much more frequent precipitation, limiting consistent irrigation to late July and early August. The flux impacts follow the timing of irrigation triggering, steadily growing throughout the summer in 2012, up to

200 W m$^{-2}$, and emerging during dry down periods in 2014. Sharp decreases in flux impacts in the time series are the result



of individual precipitation events, as the soil is not dry enough to trigger irrigation during and immediately following heavy rainfall events. In 2012, the SPoRT GVF is greater than climatology in June, resulting in more water applied and greater flux impacts in SPoRT than Tuned or Standard early in the season. However, in September, the SPoRT GVF detects the (negative) vegetation response to the July drought and irrigation amount and flux impacts are reduced. These seasonal scale

impacts illustrate that the NLDAS-2 forcing (e.g. precipitation) data, via changes to soil moisture, drives the irrigation timing during the growing season and that the behavior of the irrigation scheme is consistent with expectations of human triggering of irrigation during dry and wet periods.

### 4.3 Model sensitivity at the diurnal scale

At the interannual and seasonal scale, irrigation amounts and impacts are driven primarily by background rainfall regime,

given by the forcing precipitation, with only small changes evident between the methods. At the diurnal scale, however, the choice of greenness and irrigation intensity datasets becomes more influential to irrigation impacts. Figure 6 shows the change from control in domain average latent heat flux for each of the irrigation runs for three diurnal cycles in July 2012 and the differences from control in latent heat flux at noon, spatially. All irrigation runs result in large increases to the latent heat flux, but while Tuned and Standard are relatively close in magnitude, the SPoRT run increases latent heat flux by more

than 100 W m$^{-2}$ more than Standard during peak heating. Spatially, the SPoRT simulation has a larger change from control everywhere in the domain as compared to Standard and Tuned, which exhibit similar magnitude of differences and spatial heterogeneity.  The impacts on surface fluxes indicate that the choice of dataset, especially GVF, will likely impact coupled simulations, such as those with LIS-WRF.

In summary, the landcover, GVF, soil texture, meteorological forcing, irrigation fraction, and crop type all influence

irrigation amounts in ways that are physically consistent with expectations for crop water use. For example, it is expected that the irrigation requirement is greatest for densely irrigated areas of mature crops with dry soil; the model reproduces this scenario by applying the greatest amount of water to gridcells that have high GVF (i.e., more mature crops and deeper roots), low soil moisture (from lack of precipitation), and high irrigation intensity.





### 4.4 Model performance

### 4.4.1 Evaluation of irrigation amounts and CRNP soil moisture evaluation

The simulation of irrigation amounts and timing as well as impacts on soil moisture are evaluated for the growing season of 2014 using field observations near Waco, Nebraska, as described in Sect. 2.2. Figure 7 shows daily irrigation and rainfall

amounts (right axis), as well as the volumetric soil water content (left axis) from the in-situ CRNP (solid black line) and all model simulations (green lines) at the rainfed and irrigated maize sites. The precipitation data confirm that 2014 was a relatively wet growing season, as was originally noted in the examination of Fig. 5b. The soil at the rainfed site gradually dries out between July 15 and August 5, the only consistent rain-free period of the summer (Fig. 7a). The dry down timing is simulated well in the Control and Tuned simulations, as irrigation is not included in Control and is prohibited at the rainfed

site in Tuned, as defined by the edited irrigation intensity map (i.e., 0% for this gridcell). In contrast, the Standard and SPoRT simulations consider the rainfed gridcell to be 100% irrigated, as given by the GRIPC dataset, and as a result, both runs incorrectly trigger irrigation at this site, increasing SM during the dry down period.

At the irrigated maize site, irrigation is applied during the rain-free period in mid- July and early August and during a second, shorter stint late in August (red bars, Fig. 7b). The model simulations generally overestimate the amount of irrigation

water at the irrigated site, applying an average of 8-15 mm day$^{-1}$(not shown), while the observations show that the irrigated field generally received 5 mm day$^{-1}$ .  In contrast to the rainfed site, the CRNP observations show SM increases or remains steady in mid-July through early August due to irrigation by the farmer at the maize site.

The triggering of irrigation during the dry down period is simulated well by the model as evidenced by the soil moisture differences between the Control and irrigated runs at the irrigated maize site (i.e. dry down versus steady SM levels,

respectively). The SM given by the irrigated simulations matches the CRNP observations more closely than Control during the dry down period. This indicates that the combination of NLDAS-2 forcing and the triggering thresholds are sufficient to activate irrigation during rain-free periods, even in a wet year. Each irrigated LIS simulation applies enough irrigation water to maintain the SM levels, with small but inconsequential variations in the day to day to variability.

The soil water content observations are consistently greater than that of the model at both the rainfed and irrigated sites.

However, it is common for soil moisture probes, other observations (e.g., satellite) and land surface models to exhibit





different soil moisture climatologies that are largely a function of different representative depths of the soil (e.g. in model vs. CRNP). The spikes in soil moisture shown in the probe observations are represented well by the model, once again indicating the accuracy of the NLDAS-2 meteorological forcing data, even at this local scale. Overall, these results show that the irrigation scheme simulates well the irrigated versus rainfed soil moisture differences when the irrigation location is

specified properly by the irrigation intensity dataset (in this case, the Tuned simulation).

### 4.4.2 Evaluation with CRNP gridded product

In order to assess whether soil moisture heterogeneity due to irrigation across the domain is captured accurately, simulations are evaluated against the CRNP gridded soil moisture product. The gridded product from Franz et al. (2015) uses the spatiotemporal statistics of the observed soil moisture fields, as obtained via the CRNP rover, and a spatial regression

technique to create a 1-km, 8-hour gridded soil moisture product for the growing season (May – Sept, 388 values). In this study, we modify the spatial regression technique to treat irrigated and non-irrigated areas differently by using the CRNP (irrigated) rainfed data in the regression for (irrigated) non-irrigated gridcells. This results in a gridded soil moisture product that retains the spatiotemporal differences of the rainfed and irrigated areas.

The LIS-simulated soil moisture variability is evaluated in time and space using a comparison of the cumulative distribution

functions (CDFs) generated from the LIS simulations and the modified COSMOS product, shown in Figures 8-9. Analyzed first is the CDF of all soil moisture values in the domain for two separate days, July 25 and July 30, during which irrigation was applied at the irrigated maize site (Fig. 8). As this CDF provides information about the variability of soil moisture spatially in the study area at one particular time, it is hereafter referred to as a 'spatial CDF' (Fig. 8). Also examined is a CDF of the domain-averaged soil moisture values from May 5 to Sept 22 at 8-hour intervals (the same as the COSMOS

product; 388 values), hereafter referred to as the 'temporal CDF' (Fig. 9).

The spatial CDFs (Figs 8a-b) show uniformly dry soil in the control simulation while the irrigated runs exhibit a step-like behavior as a result of irrigation triggering and dry down timing across the domain. The different levels of steps within the irrigated simulations are a result of the input parameter datasets, as triggering and timing are dependent on these datasets The model distributions do not match the CRNP CDF, which instead shows a majority of soil moisture values that are wetter than

the control simulation but drier than the irrigated simulations and exhibit a smoother distribution. These CDFs suggest that



the model, even with the irrigation algorithm turned on, is not able to accurately simulate the small-scale (i.e. field scale) heterogeneity in soil moisture that is present in the CRNP data. The heterogeneity at this time and space scale results from the fact that center pivot irrigation systems typically take about three days to complete one rotation, so that the most recently treated slice of the field is always wetter than the rest. Further, individual decisions made by farmers on and immediately

preceding this date (USDA 2014) are not captured by the strict soil moisture deficit based rules imposed by the irrigation algorithm, nor by the uniform land cover, soil type, and slowly varying GVF datasets at 1km resolution.

In contrast, the bulk temporal variability in soil moisture in both irrigated and non-irrigated areas during the growing season is simulated well by the model (Fig. 9). The temporal CDF shows that the model matches the CRNP distribution more closely when the irrigation algorithm is turned on (Fig. 9a). Furthermore, when irrigated and non-irrigated areas are averaged

separately, the irrigated (Control) simulations match the distribution of irrigated (non-irrigated) areas well (Fig. 9b). These results suggest that if this domain were one gridcell in a larger, coarser resolution domain (e.g. 15 km spatial resolution), the variation in the gridcell soil moisture (given here by the domain average) over the growing season would be representative of observations. That is, the heterogeneity and smaller scale processes resolved in the high-resolution domain, though unable to reproduce specific field-scale behavior, appropriately scale up to coarser resolution. At coarser time and space resolutions,

the decisions made by individual farmers become less important, in favor of the larger scale features (e.g. timing of precipitation during the growing season), that influence and drive the collective behavior of human practices in this region.

**5. Discussion**

Although the responses of the modeled states and fluxes to simulated irrigation will vary depending on the LSM and irrigation scheme used, the results of this study are broadly relevant to irrigation modeling development as a whole. In

particular, this study demonstrates the importance of supplying a land surface model with high-quality input datasets. Of primary importance are the datasets that control irrigation triggering (e.g., landcover, meteorological forcing, irrigated area), as the details of irrigation application are relevant only after irrigation is triggered at the proper locations and at the correct times during the season. Once reasonable timing and placement have been established, the datasets that regulate the amount of water applied (e.g., irrigation intensity, root depth, GVF) become important. These datasets may require a certain degree





of customization, depending on the available information about irrigation practices, water district regulations, and land use in the study area, to ensure an appropriate amount of water is applied.

The root systems of crops generally mirror the vegetative state above ground (i.e., GVF), and as such, the model represents root growth by scaling the maximum root depth by the GVF (Ozdogan et al. 2010) and applying a proportional amount of

irrigation water. Although the crop type is uniform maize for the limited domain, as given by the UMD crop dataset, Franz et al. (2015) shows a mix of maize and soybeans in the study area. An additional run was completed in which a tuned crop type map was supplied to the model to distinguish between maize and soybean gridcells based on the land use map of Franz et al. (2015) and the maximum root depth was altered to be 1.2 m for maize and 1 meters for soybean. The results of this analysis showed very little differences between this simulation and the others, indicating that the model is quite insensitive to the

maximum root depth change and that the scaling by GVF tends to be more important than small changes (up to 20% in this case) in maximum root depth. However, models that contain a more complex treatment of crops may have a greater dependency on crop root depth.

The method for determining the start and end of the growing season, based on the 40% annual range in climatological GVF, proved to be reliable for this study area and climate. However, in arid or semi-arid regions, the 40% threshold applied to a

small annual range in GVF can result in a year round irrigation season that may not be representative of regional irrigation practices. Thus, where the annual range in GVF is small (e.g., southern California), more tailoring may be needed to ensure that irrigation occurs only during the local irrigation season.

This study shows model sensitivity to the irrigation intensity dataset, in terms of where and how much irrigation water is applied. Historically, the Global Map of Irrigated Areas (GMIA; Döll and Siebert, 1999) has been the most widely used

irrigation dataset in irrigation modeling studies (Bonfils and Lobell, 2007; Boucher et al., 2004; Guimberteau et al. 2012; among many others) as it was the first reliable global irrigation map, making use of cartographic and FAO statistics. However, progress in satellite remote sensing and ease of access to required datasets will likely result in a growing number of options for irrigation intensity datasets in the coming years. As such, the results of this study, detailing the potential effects of choice of irrigation intensity dataset on irrigation amounts will likely become more relevant with the expansion in

choices of irrigation-related datasets.





## 6. Conclusions

This study provided an assessment of the sprinkler irrigation physics and model sensitivity to irrigation intensity and GVF datasets in a LSM framework, and evaluated the results with novel point and gridded soil moisture observations. As expected, model results show that irrigation increases soil moisture and latent heat flux, and decreases sensible heat flux.

Differences between experiments with different GVF and irrigation intensity parameters are small at large and interannual scales, but become more substantial at small and subseasonal scales. The irrigation scheme uses GVF as a proxy for plant maturity and scales the amount of water applied accordingly to represent differences in irrigation scheduling based on growth stage. This behavior and the impacts of irrigation on soil moisture and fluxes are physically consistent with expectations of irrigation effects on the land surface.

The evaluation with CRNP observations revealed both limitations and strengths of the irrigation algorithm. Field-scale heterogeneity resulting from the slow rotation rates of center pivot irrigation systems and the individual actions of farmers are not captured by the model. Also, the amount of irrigation applied by the model exceeds that applied at the two irrigated fields. However, the timing of irrigation during the growing season (i.e., late July to early August), which coincided with a stretch of limited rainfall, is simulated well by the scheme. Additionally, the fine scale processes resolved in the small

domain appropriately scale up in time and space, indicating the scheme could be used reliably at coarser resolution (e.g. 15 km) in this region. The model skill is due in large part to the accuracy of NLDAS-2 meteorological forcing, land cover, and irrigation intensity datasets, which are all critical to reproducing the seasonal timing and location of irrigation triggering. Overall, these results underscore the importance of supplying a LSM with high-quality input datasets.

This study has also shown that CRNP distributed soil moisture data can be valuable in LSM and irrigation parameterization

evaluation. The CRNP observations provide valuable information about the impact of irrigation on the spatial and temporal variability of soil moisture, and could possibly be used to help identify where and when irrigation occurs. Irrigation timing information is particularly valuable at the scales of this study and larger, where accurate reporting data are not always available. The USDA census of agriculture contains some of the most detailed information on the state of agriculture in the U.S., including estimates of irrigated acreage, irrigation method, and crop cultivated. However, the census occurs only once

every five years and lacks irrigation timing information. CRNP soil moisture could potentially be used to fill those data gaps.





It is logical that satellite based soil moisture and evapotranspiration would also help in that respect, although a recent study cast doubt on the utility of the former (Kumar et al. 2015)

The flexibility of the LIS framework, and in particular the ability for the user to choose the irrigation scheme, parameters, and model of choice, makes LIS a premiere framework for irrigation studies. However, the general conclusions of this study,

as they pertain to irrigation scheme impacts and sensitivity to dataset changes, are applicable to irrigation modeling more broadly. The continued evaluation and improvement of irrigation parameterizations, as discussed here, is an important step towards better understanding human influences on the water cycle and the impacts of such activities in a changing climate.

**7. Data Availability**

Fixed and mobile cosmic-ray neutron probe data is available in Franz et al. (2015) or by request from Trenton Franz.

**8. Acknowledgements**

TEF would like to thank the Daugherty Water for Food Global Institute and the Cold Regions Research and Engineering Laboratory through the Great Plains CESU for financial support. TEF would like to thank Chase Johnson, Romohr Farms, and the residents of Waco, NE for access to field sites, data, and patience for slow rover driving. This work was partially supported by the NASA Earth and Space Science Fellowship.

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





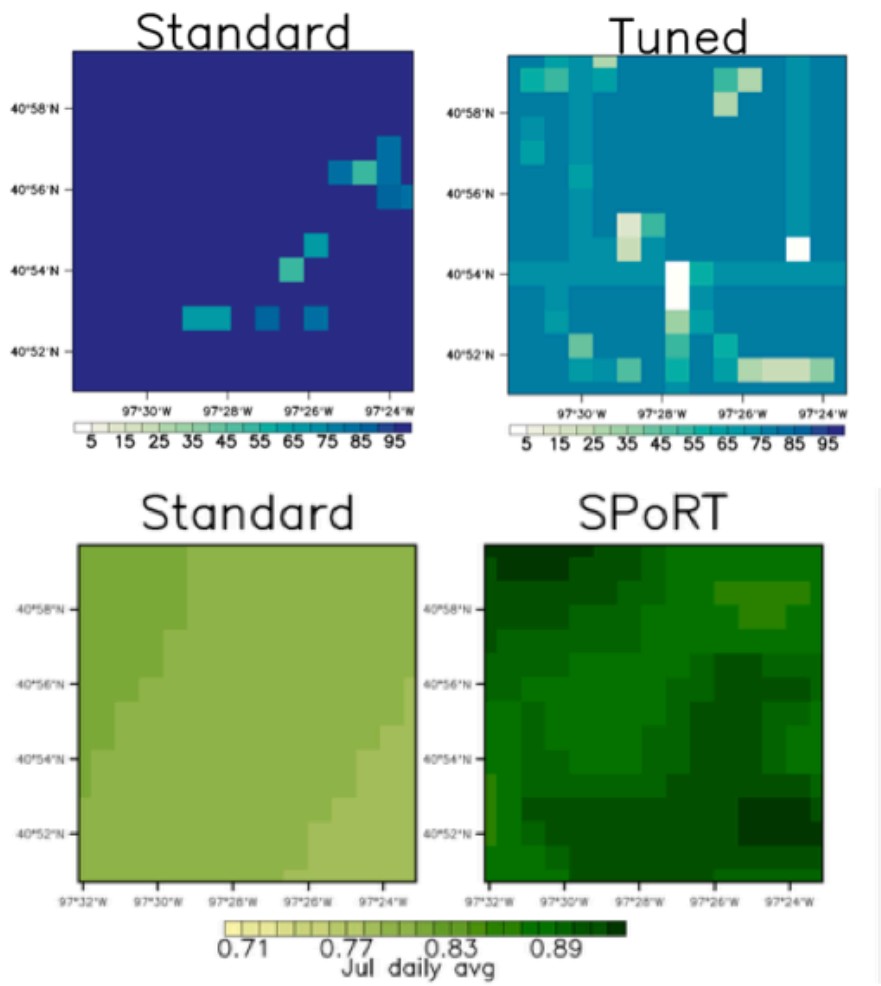

**Figure 1: Comparison of the GRIPC irrigation intensity given by Salmon et al. (2015, top left) used in the Standard and SPoRT simulations and the observationally tuned irrigation intensity (top right) used in the Tuned simulation. Average July 2012 greenness vegetation fraction given by NCEP climatology (bottom left) used in the Standard and Tuned simulations and SPoRT real-time dataset used in the SPoRT run (bottom right).**





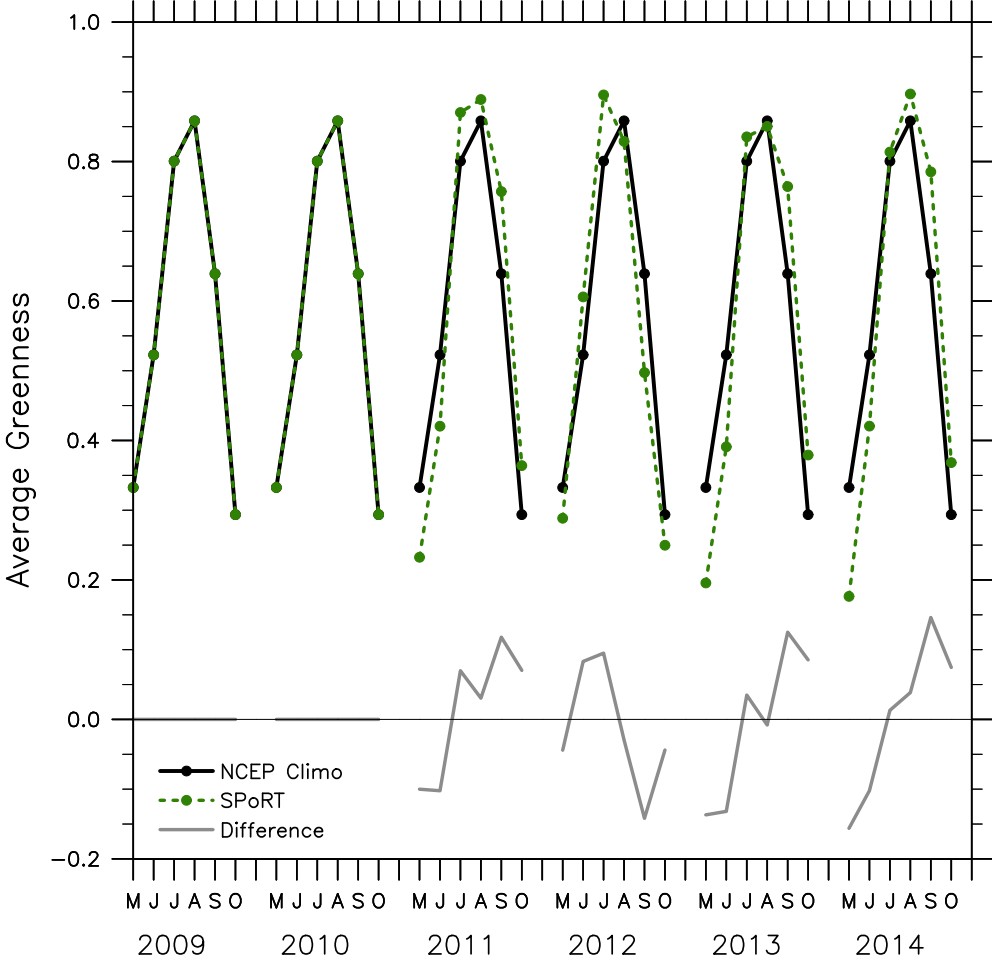

**Figure 2. Domain and monthly averaged GVF from the NCEP climatological GVF dataset, used in the Standard run, the SPoRT GVF dataset used in the SPoRT run, and the difference between the two (SPoRT – Climatology).**




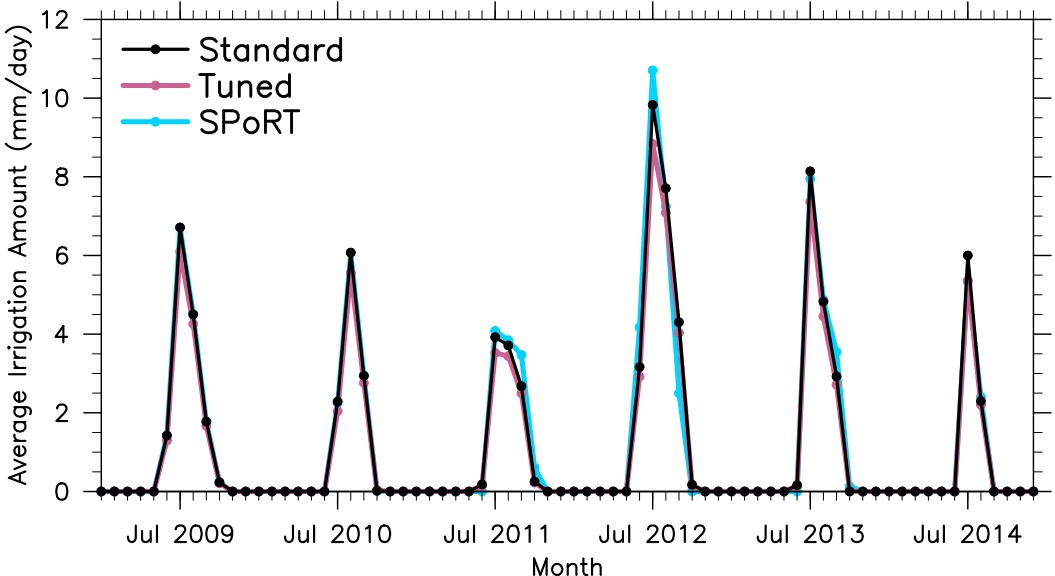

**Figure 3.** Domain and monthly averaged irrigation amount for each irrigation simulation.

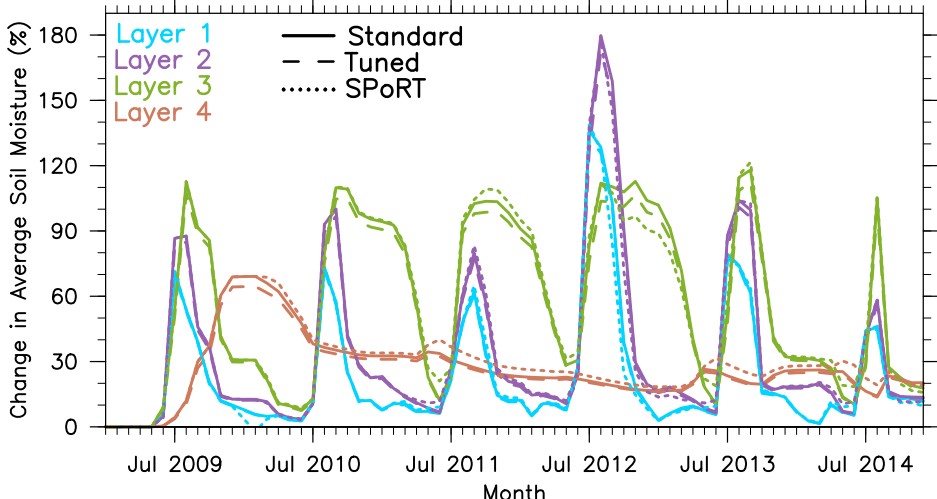





**Figure 4. Change from control (IRR - CTRL) in soil moisture for each experiment (line style) and each layer (line color). Layer designations are the Noah LSM default layers Layer 1 (top layer) is 0 to 10 cm depth, layer 2 is 10 to 40 cm (delta Z = 30cm), layer 3 is 40 cm to 1 m (delta Z = 60 cm) and layer 4 is 1 m to 2 m (100 cm depth).**

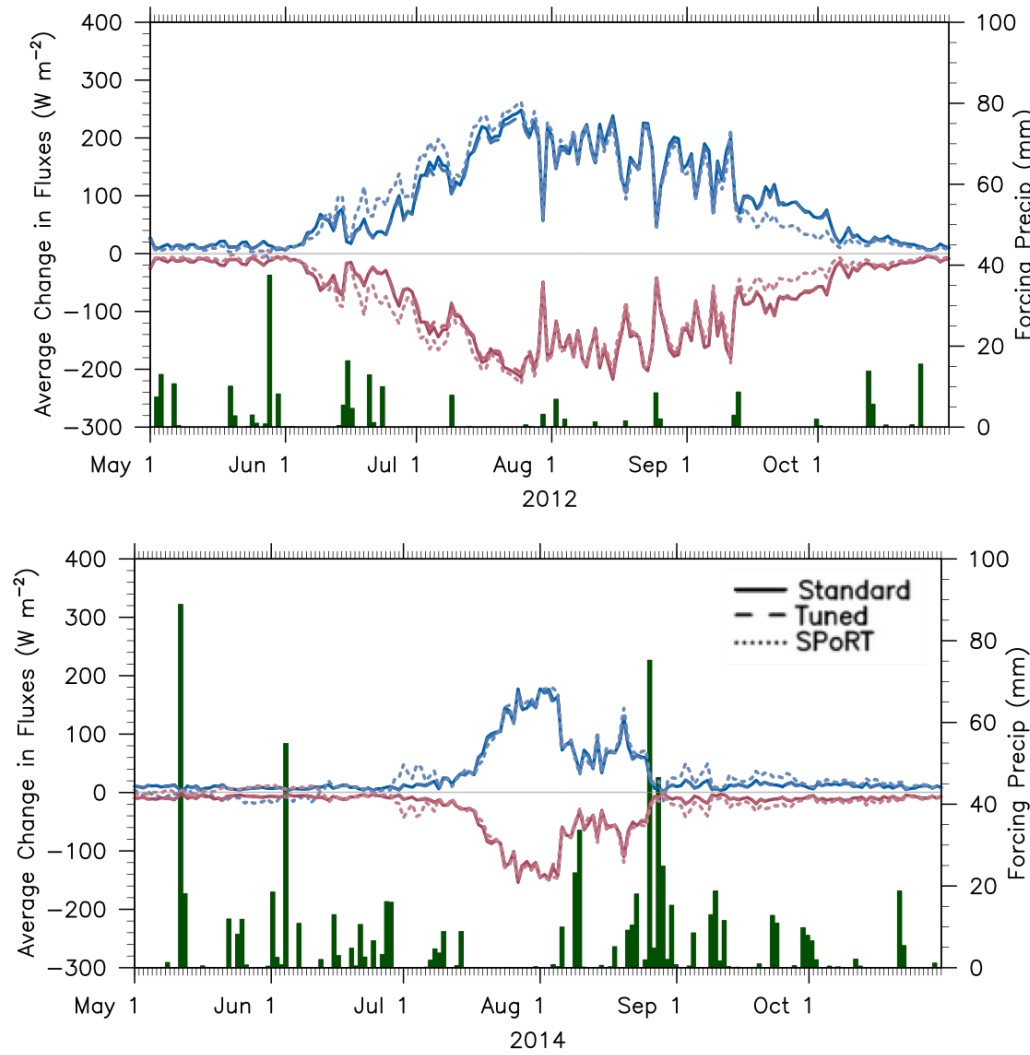

5   **Figure 5. May to September 2012 (top) and 2014 (bottom) domain average daily change from control (IRR-CTRL) in latent (blue) and sensible (red) heat fluxes for each irrigation simulation (left axis) and domain average daily accumulated precipitation from the NLDAS2 forcing data (right axis).**

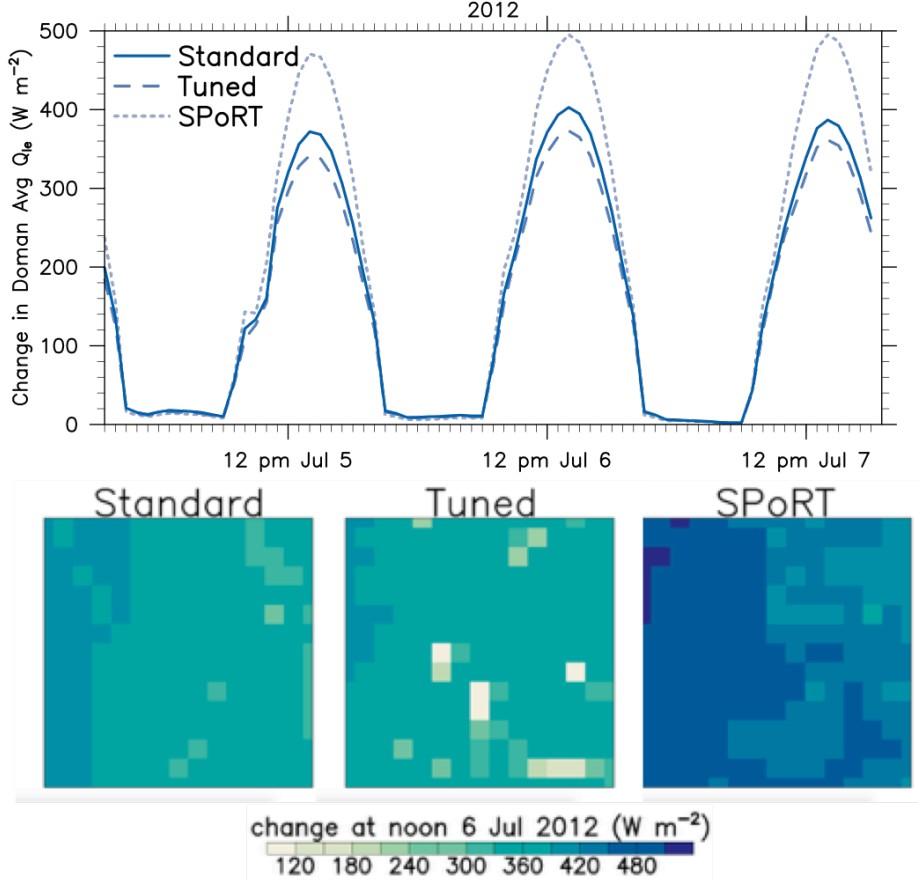

**Figure 6.** Domain average change in latent heat flux for three diurnal cycles in July 2012 (top). Change in latent heat flux (IRR-CTRL) at noon on July 6, 2012 for each irrigation simulation (bottom).





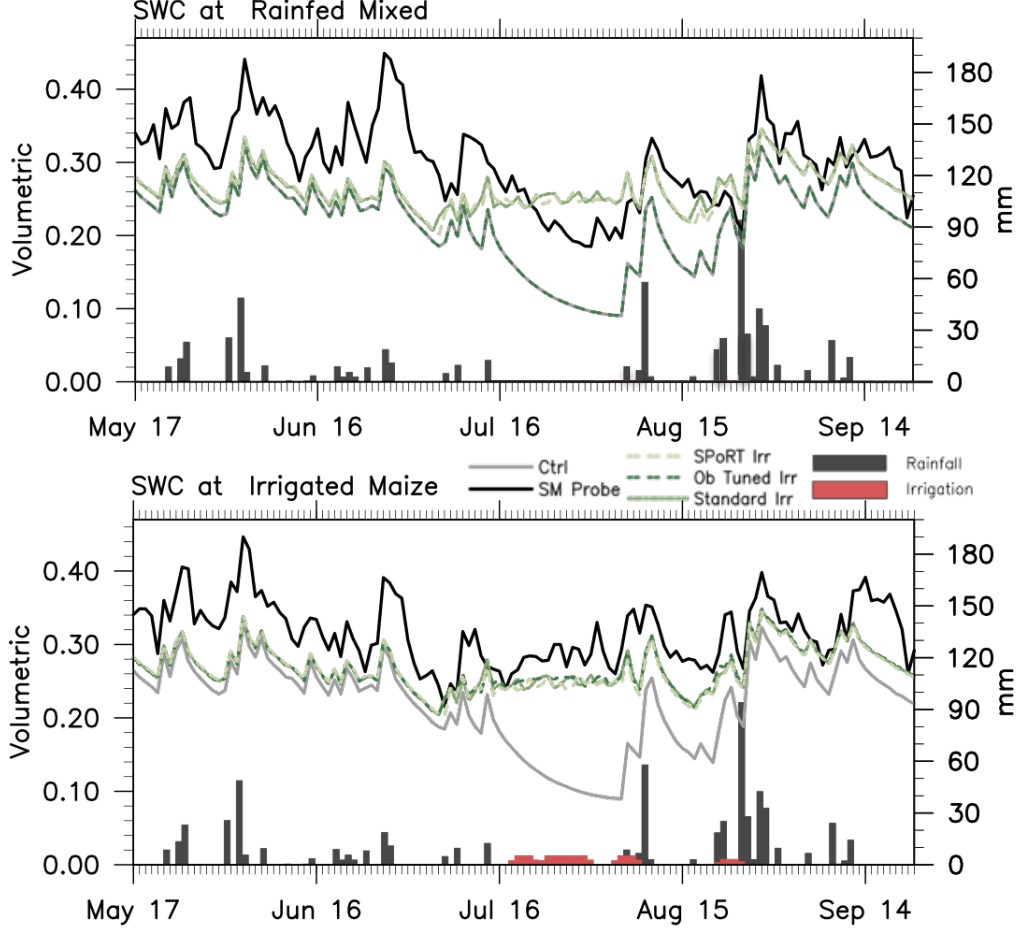

**Figure 7.** Volumetric soil water content at the rainfed (top) and irrigated maize (bottom) sites (left axis). The black solid line shows observations from the CRNP probe, the gray and green lines show the LIS control and irrigation simulations, respectively. Dark gray bars show accumulated daily precipitation from the Automated Daily Weather Network in York, Nebraska and pink bars show the accumulated irrigation amount at the irrigated maize and soybean sites (right axis).





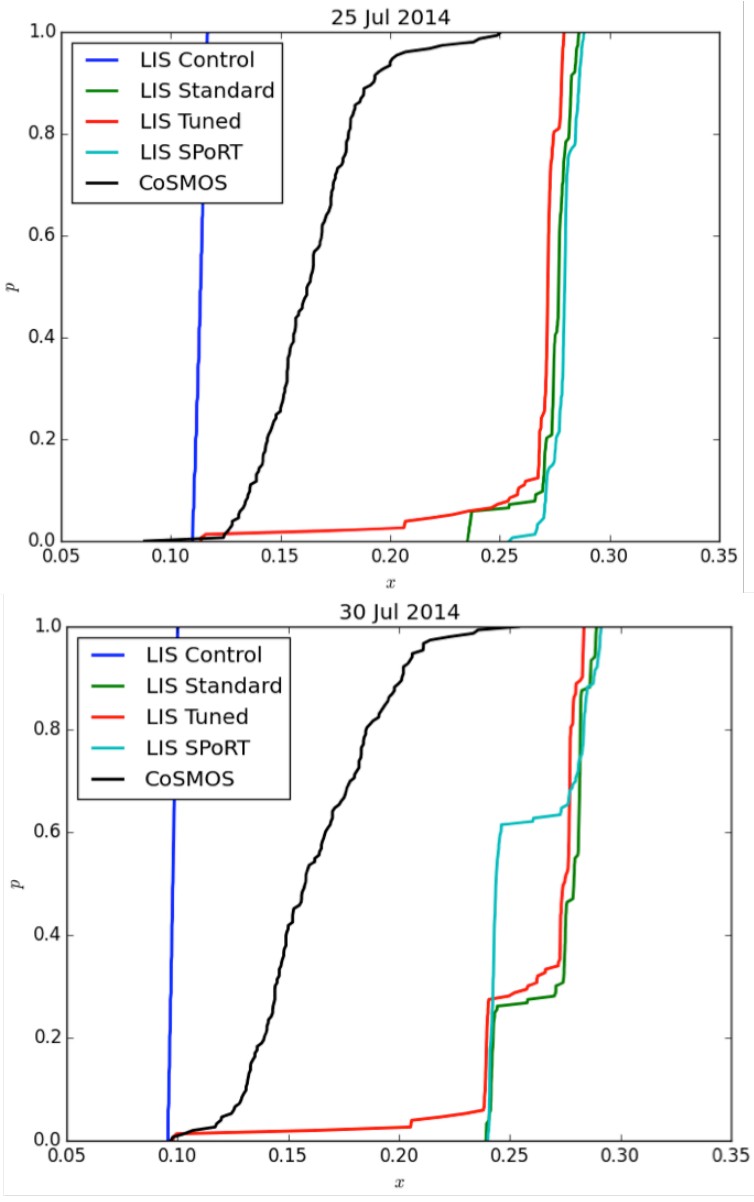

**Figure 8. Spatial CDF for 25 July 2014 and 30 Jul 2014, two dates when irrigation was applied at the irrigated maize and soybean sites in practice and in the model simulations.**





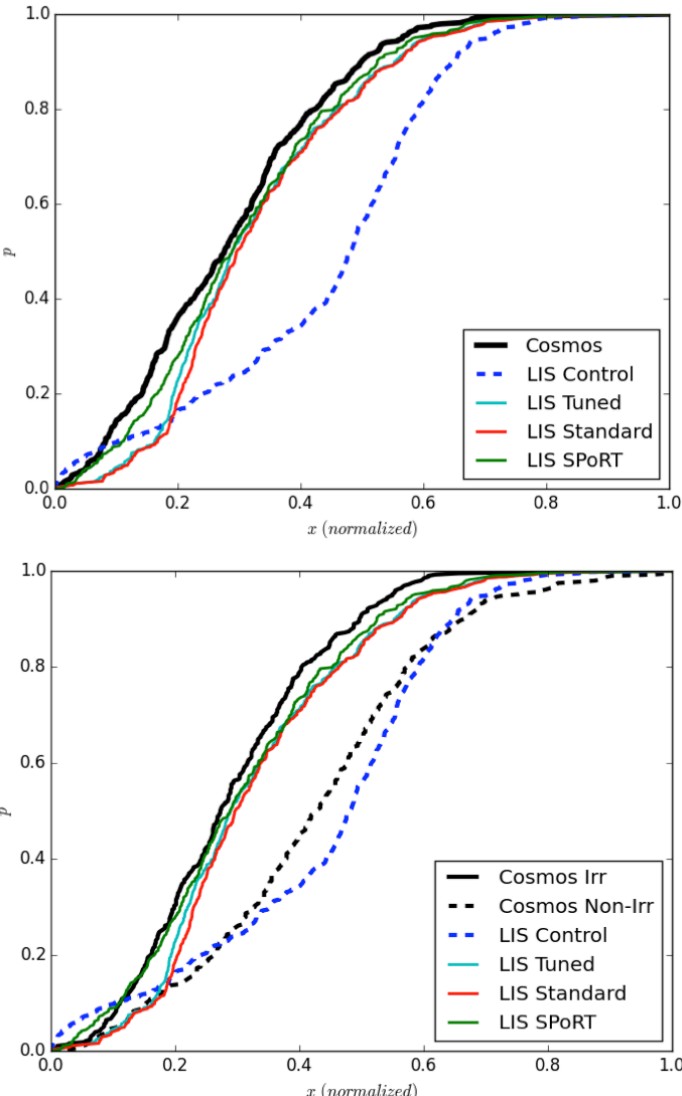

**Figure 9. Temporal CDF of normalized domain averaged (top) and irrigated/non-irrigated spatial average (bottom) SWC values from May 5 to Sept 16 from the COSMOS observational product (black) and the model simulations (colors).**

