# Peer review of "Assessment of Irrigation Physics in a Land Surface Modeling Framework using Non-Traditional and Human-Practice Datasets"

_Hydrology and Earth System Sciences, 2017_

## Referee Comment (RC1) · O. Lopez (Referee) · 28 Feb 2017

**1   Summary**

The manuscript presents a study of the impacts that a sprinkler irrigation scheme in a land surface model have on the latent and sensible heat fluxes, and more substantially in the soil moisture state on a small, high resolution domain containing center-pivot sprinkler irrigation systems. The study explores the sensitivity of the results to two parameters: the irrigation intensity as prescribed by an input data set (GRIPC), and the greenness vegetation factor (GVF) used to scale the irrigation amount depending on the growth stage of the crops. The soil moisture state is compared to fixed soil

moisture probes and a gridded soil moisture product, both using Cosmic Ray Neutron Probes. Including irrigation in land surface models is becoming more important to properly characterize the state and fluxes in agricultural regions, and thus efforts to evaluate the impact that either the choice of irrigation scheme or their input datasets have on the model results is certainly relevant to HESS. The study introduced modifications to the irrigation scheme such as using a real-time greenness vegetation factor data (as opposed to a climatological one) and also introduced a modification in the method to develop a soil moisture gridded product.

Overall, the manuscript is well written and the conclusions reached are sufficiently supported by the results. However, there are some few comments that I think would improve the readability of the manuscript, particularly with the description of some of the input datasets (GRIPC and in situ irrigation) as well as part of the methodology. Therefore, my recommendation is acceptance with minor revision.

**2   General comments**

1. The title refers to *"non-traditional"* and *"human-practice"* datasets. However, it is not clear what the authors mean by these two concepts. It might be the case that *"non-traditional"* is referring to the use of Cosmic Ray Neutron probes, but this is not obvious. In contrast, "human-practice" data is defined in Page 6, line 13 to be the irrigation amount. However, it is not clear if this term is referring to the GRIPC dataset used throughout the study (which is based not only on human data, but also on remote sensing data), or to the amount of irrigation applied at two sites, as mentioned in Page 6, line 10.

2. Related to the previous comment: although a reference is given for the GRIPC dataset, a brief description of this dataset would benefit the manuscript. An estimate of the uncertainties related to this dataset would also be helpful.

3. Also related to the first comment: a description of the irrigation data from the study in Franz et al. (2015) is also worth including. This is especially important in Figure 7, where irrigation at the maize site is shown, as well as in the text (Page 12, lines 14-16).

4. The methodology for defining the growing season was not included in the Methods section. It is however mentioned later in the Discussion section on Page 15, lines 13-14 *"The method for determining the start and end of the growing season, based on the 40% annual range in climatological GVF, proved to be reliable for this study area and climate"*.

**3  Minor comments**

1. Page 2, line 9: (referring to observational data) *"are generally not obtainable at the scale of LSMS"* and Page 5, lines 19-21: *"available at the same spatial scale as LSMs"*

What do the authors mean by scale of LSMs? land surface models can be run at a great range of scales. Perhaps the authors are talking specifically about high-resolution LSMs such as in this study? If so, please specify this.

2. Page 4, line 1: *"For example, a flood irrigation parameterization..."*

It is not clear if this is referring to scheme number 1 or 2 defined above in Page 3, lines 19-22. The text would benefit if this term ("flood irrigation parameterization") would be included in Page 3, lines 19-22 where applicable.

3. Figure 1: The titles in each sub-figure are confusing. Perhaps the titles could read (top left, top right, bottom left, bottom right): "GRIPC irrigation intensity", "Tuned irrigation intensity", "Climatological GVF", and "Real-time GVF" to better identify what is being shown. Furthermore, the figure would improve by the inclusion of labels "a", "b", "c" and "d". Finally, the colorbar for the top figures (which is the same for both) could be shown in the center as it was done for the bottom figures.

4. Page 11, line 2: *"the SPoRT GVF is greater than climatology in June"*

Please clarify: do the authors mean "greater than climatological GVF"?

5. Page 11, lines 3-4. *"However, in September, the SPoRT GVF detects the (negative) vegetation response to the July drought and irrigation amount and flux impacts are reduced".*

What do you mean by "the sport GVF detects the negative vegetation response to the July drought?" is it because it is a real-time product as opposed to the climatological product and the fact that 2012 was particularly dry?

6. Page 11, lines 4-7. *"These seasonal scale impacts illustrate that the NLDAS-2 forcing (e.g. precipitation) data, via changes to soil moisture, drives the irrigation timing during the growing season and that the behavior of the irrigation scheme is consistent with expectations of human triggering of irrigation during dry and wet periods".*

I am not sure I follow completely what is meant here. Is this saying that we expect irrigation triggering when there is no (or small amounts of) precipitation and no triggering when there is? If so, then this is already phrased better in the next lines (page 11,

lines 9-10): *"At the interannual and seasonal scale, irrigation amounts and impacts are driven primarily by background rainfall regime, given by the forcing precipitation, with only small changes evident between the methods"*.

7. In Figure 7, why not include the soil moisture from the CRNP gridded soil moisture product as well for comparison with the fixed probes?

8. Page 13, lines 11-12 *"In this study, we modify the spatial regression technique to treat irrigated and non-irrigated areas differently by using the CRNP (irrigated) rainfed data in the regression for (irrigated) non-irrigated gridcells"*.

I am not sure I follow the last part with the parentheses "by using the CRNP (irrigated) rainfed data in the regresion for (irrigated) non-irrigated gridcells". Could you please clarify this?

9. Referring to the same text in the last comment, in my opinion, since this is also a novel contribution (the modification of the spatial regression technique for the gridded product), a comparison between the previous and the new product could be included as supplementary material.

10. Page 13, lines 17: *"during which irrigation was applied at the irrigated maize site"*

Only at the maize site? or the whole domain shown in Figure 1? The caption reads *"when irrigation was applied at the irrigated maize and soybean sites"*. To my understanding, the maize site and soybean sites are only parts of the whole domain, and this figure (Figure 8) is showing a spatial comparison of the whole domain.

11. Figure 8: In the legend, consider changing "CoSMOS" to "CNRP" to be consistent with the rest of the paper.

12. Page 14, lines 9-10 *"Furthermore, when irrigated and non-irrigated areas are averaged separately, the irrigated (Control) simulations match the distribution of irrigated (non-irrigated) areas well"*.

Again, I do not understand the use of the parenthesis here "irrigated (non-irrigated)".

**4  Technical corrections**

1. Page 7, line 19 *"as evidenced by only 5% of the gridcells having intensity less than 100% (Fig 1)"*

I think this should be "Fig 1a" instead of "Fig 1".

2. Page 7, line 22 *"(i.e. observationally tuned: Fig 1)"*

I think this should be "Fig 1b" instead of "Fig 1".

3. Page 7, lines 19 and 22 and Page 8, line 4.

Check consistent use of either "(Fig X)" or "(Figure X)".

4. Figure 6: Label in Y-axis "Change in Domain Avg Qle" instead of "Doman"

5. There is a dot missing in Page 13, line 24 before "The model distributions do not match the CRNP CDF, which instead shows..."

---

## Referee Comment (RC2) · Anonymous Referee #2 · 10 Mar 2017

Summary: The authors provide a useful and clearly-written evaluation of irrigation simulated by an advanced Land Surface Model. These types of evaluation are in short supply, and the use of CRNP in model evaluation is, to my knowledge, novel and potentially quite useful. I believe that the Discussion Paper is of sufficient interest and quality for publication in HESS. That said, the numerical experiments presented in the study are rather limited. Sensitivity to GVF dataset and irrigation intensity factor are evaluated, but none of the many other factors that the authors list are explored. This may lead to the wrong impression that the tested factors are the most important when simulating irrigation, when I see no evidence presented by the authors that this is in fact the case. Ideally, the authors should present a more inclusive set of sensitivity tests

to inform future modeling studies about the relative importance of different factors. If this is not possible, or if the authors view it as unnecessary, then a more convincing justification for the choice of experiments is required.

General Comments:

1. Meteorological Forcing: In the abstract and at several other passages in the text the authors emphasize the importance of high quality meteorological forcing data for accurate simulation of irrigation. Their results suggest that NLDAS is high quality, as shown most convincingly by the temporal match of simulated irrigation to spikes in observed soil moisture. I believe that NLDAS is high quality and that these results show impressive performance at local scale. But I'm not sure that the authors can actually make any conclusions about the importance of forcing data to irrigation simulations, given that they do not compare NLDAS simulations to simulations with any lower quality forcing dataset. Yes, it its intuitive that simulations with NLDAS will be better, but the numerical experiments don't demonstrate this, and they don't show us *how* important it is. This is particularly the case when one considers spatial or temporal scale. The authors nicely demonstrate that simulations are more realistic at larger and longer scales than they are at local and shorter scales. How important is meteorological forcing if we are concerned with large and long time scales? Additional simulations with an alternative, poorer quality meteorological forcing dataset would be the obvious way to test this, but the authors might find other ways to make the point.

2. Thresholds: The authors appropriately emphasize the importance of selecting proper thresholds for soil moisture and GVF at several points in the text. But the manuscript does not offer any evaluation of either. In both cases a single threshold is applied and attributed to previous studies. It would be quite interesting to know how the impact of using different GVF datasets compares to differences caused by small changes in GVF threshold. And how does a modest change in threshold impact total water use, as compared to the tested sensitivity to prescribed irrigation intensity? I understand that no study can be comprehensive on all parameters, but I don't fully

understand why the authors chose to look only at GVF dataset in GRIPC irrigation intensity when other subjective modeling decisions might have as large or larger impacts on the simulations. If possible I would encourage the authors to expand their sensitivity test in order to justify the selection of these two factors as the focus of study.

Minor Comments:

Page 3, line 20: This list of options misses flood irrigation simulation (unless it's supposed to be covered by #1). Several studies have employed flood irrigation, including Yilmaz et al. (2014), Leng, and Evans & Zaitchik (2008).

Section 2.3: It would be useful to include a sentence or two on why CRNP measurements are sensitive to soil moisture. Many readers (myself included) are not deeply familiar with this technique.

---

## Referee Comment (RC3) · Anonymous Referee #3 · 17 Mar 2017

I. Summary

This manuscript examines the issue of developing and validating realistic irrigation schemes for use in land surface models (LSMs). In this study, the authors utilize observation-based datasets of irrigation intensity and green vegetation fraction (GVF) to tune the LSM irrigation amounts, which are validated against data obtained from Cosmic Ray Neutron Probes (CRNP). The main conclusion of the authors is that the timing, amount, and spatial spread of irrigation are more sensitive to the choice of irrigation scheme at smaller spatiotemporal scales than at larger, more typical scales for regional climate models. Given the balance of evidence presented and the use of a novel dataset (CRNP) for addressing this issue, it seems that the authors have arrived

at robust and meaningful conclusions that would be worthwhile additions to HESS and to the field of hydrology, in general. While I have no major qualms with the content or substance of the manuscript, I do present below some more minor comments for improving the robustness and presentation of the results.

II. General comments

A. NLDAS-2 – The authors mention several times throughout the manuscript the need for "high-quality" meteorological forcing and point out repeatedly the accuracy of the precipitation data from NLDAS-2 for their domain. While it certainly seems that NLDAS-2 provides accurate forcing over this domain (and is a high-quality dataset, in general), I echo Reviewer 2 in cautioning against drawing far-reaching conclusions about NLDAS-2 from this limited study. The entire study domain is 15 x 15 km, very small even for typical regional climate model simulations; the entire domain would fit in 4 grid cells of NLDAS-2 (1/8 degree horizontal resolution). Is there evidence that NLDAS-2 would provide equally accurate data for a different domain within the same region, or in a different region or year? If so, then I would provide a sentence or two explaining the skill of NLDAS-2 over the general region (e.g., Great Plains/Midwest) during the growing season (perhaps from the Xia study). If not, then please temper the language emphasizing the high quality of NLDAS-2 with the understanding that the spatial domain of this study is extremely limited and that NLDAS-2 may not be as accurate in other agricultural regions in North America.

III. Specific comments

A. Page 14, line 10 – "These results suggest that if this domain were one gridcell in a larger, coarser resolution domain (e.g. 15 km spatial resolution), the variation in the gridcell soil moisture (given here by the domain average) over the growing season would be representative of observations."

It would be interesting to see a supplemental model analysis with coarser-resolution grid cells (either in this paper or a future one) that validates this hypothesis. For example, what is the spatial threshold at which large-scale forcings begin to dominate the changes in the soil moisture signal?

B. Page 15, line 9 – "...indicating that the model is quite insensitive to the maximum root depth change..."

Some common irrigated crops, such as alfalfa, have max root depths of 2+ meters. Though irrigated alfalfa is much less common in Nebraska when compared to corn and soybeans, it would be instructive to not make the above claim about the insensitivity of the model to max root depths unless other crops with much larger or smaller max root depths have been tested.

C. Page 15, line 22 – "...a growing number of options for irrigation intensity datasets in the coming years".

A new global irrigation dataset (the Historical Irrigation Dataset) was published through HESS rather recently (S. Siebert, M. Kummu, M. Porkka, P. Döll, N. Ramankutty, and B. R. Scanlon (2015), "A global dataset of the extent of irrigated land from 1900 to 2005," Hydrology and Earth System Sciences. DOI: 10.5194/hess-19-1521-2015). It may deserve a citation here because of its recent development and global coverage.

D. Figure 1 – Are the spotty areas of low irrigation intensity in the Tuned plot over urban areas? A brief explanation of this in the text may be warranted.

E. Figure 2 – It would be helpful to mention in the figure caption that SPoRT uses the climatological GVF in years 2009 and 2010 (as is already mentioned in the text) to avoid confusion.

F. Figure 4 – I don't believe that IRR was ever defined (in either the main text or the figure caption).

G. Figure 4 – The boundaries of Layer 4's soil depths are only mentioned here, not in the main text. Since crop roots barely extend into this layer (max root depths of 1 or 1.2 m), perhaps this further explains why there seems to be much more variability in

soil moisture between irrigation simulations in Layer 3 than in Layer 4.

H. Figure 8 – I think that the presentation of "spatial" CDFs in this figure is rather non-intuitive. To me, it would be much more intuitive to see the differences in the spatial distributions of soil moisture within the domain using a histogram, especially since each CDF is plotted for only a single time step and thus there is no "accumulation" of data over time. In this figure, since data is accumulated spatially (in two dimensions) rather than temporally (in one dimension), the shape of the CDF would be rather arbitrary and would partly depend on the order in which you spatially sample the grid cells.

I. Figure 8 – Neither the figure nor the figure caption explain what is being plotted in the figure. Units would also be appreciated (even if unitless).

IV. Technical corrections

A. Page 1, line 17 – "at the interannual scale, but become..." – Remove the comma.

B. Page 2, line 23 – "previous evaluation efforts, and introduces..." – Remove the comma.

C. Page 3, line 14 – e.g., "de Vrese et al. 2016" – Please be consistent with placing commas after "et al." in internal citations.

D. Page 4, line 1 – "with a two different..." – Remove "a".

E. Page 4, line 2 – "in the U.S. Central Great Plains..." – "Central" should be lowercase.

F. Page 4, line 5 – "Tuinenburg et al., 2014), or in..." – Remove the comma after "2014)"

G. Page 4, line 15 – No need for commas surrounding "such as these".

H. Page 4, line 23 – "...to reproduce county and water resource region irrigation water usage..." – Change to "...to reproduce irrigation water usage within counties and water resource regions...".

I. Page 5, line 17 – Change "c.f." to "cf.".

J. Page 5, line 19 – "reliable, area-average soil water content" – Throughout the manuscript, please change to "area-averaged" or "domain-averaged" (as in the above example) when being used as an adjective and "area average" and "domain average" when being used as a noun.

K. Page 6, line 9 – Change to "Sect. 3".

L. Page 7, line 22 – "i.e. observationally tuned" – Change all instances of "i.e." and "e.g." to "i.e.," and "e.g.,".

M. Page 8, line 8 – "more sophisticated, but computationally expensive..." – Remove the comma.

N. Page 8, line 8 – "such a dynamic..." – Change to "such as".

O. Page 8, line 14 – Change to "bias-corrected".

P. Page 11, line 14 – "the SPoRT run increases latent heat flux by more than 100 W mˆ-2 more than Standard" – Change to "latent heat flux in the SPoRT run is more than 100 W mˆ-2 greater than Standard".

Q. Page 12, line 15 – Add a space between "mm dayˆ-1" and "(not shown)".

R. Page 12, line 25 – Add a comma after "(e.g., satellite)".

S. Page 13, line 11 – "CRNP (irrigated) rainfed data..." – I would discourage this parenthetical style (it already seems to have confused other reviewers). If you must use it, I would recommend putting the parenthetical expression second, e.g., "CRNP irrigated (rainfed) data". However, I would instead prefer this and related sentences to be written as: "by using the CRNP irrigated and rainfed data in the regression for irrigated and non-irrigated gridcells, respectively".

T. Page 13, line 23 – Add a period after "dependent on these datasets".

U. Page 13, line 25 – Change "exhibit" to "exhibits".
V. Page 14, line 5 – Hyphenate "deficit based".

W. Page 14, line 11 – Hyphenate "coarser-resolution".

X. Page 16, line 3 – Remove the comma after "LSM framework".

Y. Page 16, line 4 - Remove the comma after "latent heat flux".

Z. Page 16, line 21 – Remove the comma after "soil moisture".

AA. Page 16, line 23 – Change to "USDA Census of Agriculture".

BB. Page 17, line 1 – Hyphenate "satellite based".

CC. Page 17, line 2 – Add period after "(Kumar et al., 2015)".

DD. Page 17, line 4 – Change "premiere" to "premier".

EE. Page 17, line 23 – Capitalize "a" after Myhre, and ditto for all other instances of mixed case for author names in the reference list.

FF. Page 18, line 8 – Be consistent with italicizations: Either italicize all journal names or keep them all as plain text.

GG. Page 18, line 8 – Change "hess" to "HESS".

HH. Page 18, line 28 – What does "Received" mean?

II. Page 19, line 3 – Be consistent with capitalization of the article titles: Either capitalize only the first word and proper nouns (standard practice) in every title or capitalize all words in every title.

JJ. Page 20, lines 5-9 – I think that these lines are in a slightly different font than the other references.

KK. Page 21, lines 2-3 – See above comment.

LL. Page 21, line 23 – What is "Artn"? Article number?

[Figure]

MM. Fig. 1 caption – Please define the units of irrigation intensity (even if unitless).

NN. Fig. 4 caption – Add a colon after "LSM default layers".

OO. Fig. 4 caption – Be consistent with parenthetical notes: Delta Z is included for the middle layers but not for the top or bottom layers.

––––––––––––––––––––

---

## Referee Comment (RC4) · Anonymous Referee #4 · 19 Mar 2017

General comments:

This is an interesting paper on the evaluation of an irrigation scheme within a land surface modeling framework. This is an area that needs research and I see this a potentially valuable contribution on the matter.

While generally well written, the structure and organization of the Background (particularly Section 2.3) and the Methods sections needs to be improved to ensure a better flow and enhanced readability. The study region, models, input datasets and evaluations should be described in a more logical and orderly manner with less intermixing. These issues are described in more detail in the specific comments below. The discussion section is very short and would benefit from more elaboration and high quality

insights on the limitations and challenges as well as opportunities for irrigation modeling.

Some of the used input datasets need more justification. GVF is an important dataset for the irrigation modeling but is reported at coarse resolution (3 and 16 km) inconsistent with the resolution of the LSM (1 km). Not clear to me why a 1 km based version isn't used here. The MODIS phenology product (produced at 500 m resolution) would probably be more useful in this context for establishing the start and duration of the growing season.

I'm also a bit concerned that 1 km isn't the most appropriate scale to do irrigation modeling and accuracy assessments as you will inevitable run into mixing of rainfed and irrigated fields given the characteristic size of the fields. LSM runs at 500 m resolution would probably have been more appropriate, also considering the scale of the CRNP validation dataset, and feasible using widely available surface inputs generated at consistent resolutions.

Specific comments:

1) Page 1 L14: Please define the scale associated with "high resolution"

2) Page 1 L19: What precisely does the "human practice data" consist of?

3) Page 1 L21: "two irrigated fields" – what irrigated fields are you referring to here (soybean and maize)?

4) Page 2 L21 and L25: Please define what you mean by coarse and high resolution here.

5) Page 6 L1-7: This paragraph reads a bit confusing with mentioning of all the different temporal and spatial resolutions. A bit unclear what product version is used for the evaluation. Does the 12x12 km survey area correspond to the 15x15 km domain of this study? Why the domain difference?

6) Page 6 L8-16: This Section adds to the confusion by repeating some of the statements above and also adding additional evaluation datasets (human practice data etc.) not related to the CRNP (although that is the title of the Section). Differences between the CRNP and COSMOS datasets should be clarified, if any. The finishing paragraph relates the overall objectives and novelty of the work, which don't belong here. This Section requires some revision – the evaluation components might be more appropriately positioned in the method section. You may need a completely separate section for describing the additional datasets mentioned here.

7) Section 2.3: The CRNP data description is currently part of the introduction/background part of the manuscript. While it makes sense to mention and introduce the data as a useful validation source in this context, I feel that the detailed description of the actual dataset used here for evaluation purposes should be moved to a separate section in the Methods section (or Methods and data section). Here you could appropriately describe all the datasets used in the study.

8) Section 3: I would start this with a description of the study area and domain to set the stage.

9) Section 3.1: I find this section quite confusing to read as it includes both modeling and evaluation details and references to elements described in Section 3.2. I think you need to rethink the organization of the Method section adopting a more logical organization for improved flow and readability. Personally, I would prefer to have all model descriptions first before the description of experiments and evaluations to be performed.

10) Page 8 L1-5: So why isn't the GVF datasets provided at 1 km to be consistent with the LSM resolution? You also need to specify precisely what the GVF product is used for, when first introduced. From what I can read later in the manuscript it is predominantly used to determine the start and end of the growing season; couldn't you use the MODIS phenology product (see comment 12) more appropriately for this

purpose? In addition, this product is available for the full duration of the study.

11) Page 8 L12-15: You need to mention the resolution of these input datasets. Is the UMD crop type product static or is a separate classification provided for each year? The annual Cropland Data Layer (https://www.nass.usda.gov/Research_and_Science/Cropland/SARS1a.php) product (provided at 30 m) is updated for each year to account for crop rotations and changing crop type patterns and might be a more correct source to use for something like this.

12) Page 9 L4-5: The GVF product is used for establishing the length and timing of the growing season. A more appropriate source for this would be the MODIS global vegetation phenology product (MCD12Q2) currently produced at 500 m resolution that is also more consistent with the LSM resolution and the CRNP validation dataset (and the scale of irrigation effects). Reasons for not using something like this should be addressed.

13) Page 9 Section 4: A brief intro statement would be useful here.

14) Page 10 L7: The relationship used to compute the root zone length from GVF should be provided in the methodology.

15) Page 12 L6: This is the first mentioning of a rainfed validation site within the study domain. Details like this should be provided in the method section (preferably in a dedicated study region section).

16) Page 13 L8-13: This should be moved to the methodology section. A shorter summary of the CRNP would suffice here.

17) Page 13 L15: Not clear what modifications were made to the COSMOS product; provide a section reference or more details here. Also a bit confused about the references to both CRNP and COSMOS as they are presumably the same thing?

18) Page 13 L14-15: I wonder if a non-cumulative PDF wouldn't be better in this context?

19) Page 14 L6: I believe that the GVF is provided at 3 km (and 16 km) rather than 1 km resolution, correct?

20) Section 5: The discussion is very brief and lacks more substantial and high quality discussion elements on limitations, challenges and opportunities.

21) Page 15 L3-8: These are useful details that should have been provided in the methodology or result sections

22) Page 15 L9-12: Not sure I understand this correctly, particularly the part about the scaling by GVF being more important than changes in rooting depth.

23) Page 15 L13: The method for determining the start and end of the growing season hasn't been described anywhere, but it must be. Justifications for adopting that methodology (rather than relying on existing phenology products for instance) should also be provided.

Technical corrections:

1) Page 4 L1: "with a two different.." - should be "with two different.."

2) Page 4 L23: "..water resources region..."?

3) Page 5 L14: use "high resolution" rather than "high-resolution"

4) Figure 5: I would also show the irrigation amounts here as done in Figure 7. Why is the impact of irrigation high when no irrigation is applied (e.g., during rain events)?

5) Figure 5: Issue with the legends – they are not consistent with what is shown; currently I can only distinguish two different line styles.

6) Figure 5: a and b rather than top and bottom should be used for more precise figure referencing in the manuscript. This also applies to the other figures.

---

## Author Comment (AC1) · 20 Apr 2017

We would like to thank all of the reviewers for the thorough and insightful suggestions and comments. We made substantial changes to the manuscript, replaced one figure, and completed an additional model simulation in response to the feedback we received. We feel that the manuscript has improved significantly as a result of these thoughtful reviews. Please find our detailed responses to the reviewer's comments below.

Please note that the **reviewer comments are shown in black** and our author responses are in blue. Where changes have been made in the manuscript, the page and line number(s) are given. In some cases, to highlight changes to passages in the manuscript, these sections are copied and pasted from the manuscript.
* * *
**Reviewer #1: Oliver Lopez**

**Summary:**

The manuscript presents a study of the impacts that a sprinkler irrigation scheme in a land surface model have on the latent and sensible heat fluxes, and more substantially in the soil moisture state on a small, high resolution domain containing center-pivot sprinkler irrigation systems. The study explores the sensitivity of the results to two parameters: the irrigation intensity as prescribed by an input data set (GRIPC), and the greenness vegetation factor (GVF) used to scale the irrigation amount depending on the growth stage of the crops. The soil moisture state is compared to fixed soil moisture probes and a gridded soil moisture product, both using Cosmic Ray Neutron Probes. Including irrigation in land surface models is becoming more important to properly characterize the state and fluxes in agricultural regions, and thus efforts to evaluate the impact that either the choice of irrigation scheme or their input datasets have on the model results is certainly relevant to HESS. The study introduced modifications to the irrigation scheme such as using a real-time greenness vegetation factor data (as opposed to a climatological one) and also introduced a modification in the method to develop a soil moisture gridded product.

Overall, the manuscript is well written and the conclusions reached are sufficiently supported by the results. However, there are some few comments that I think would improve the readability of the manuscript, particularly with the description of some of the input datasets (GRIPC and in situ irrigation) as well as part of the methodology. Therefore, my recommendation is acceptance with minor revision.

**General comments**

1. The title refers to *"non-traditional"* and *"human-practice"* datasets. However, it is not clear what the authors mean by these two concepts. It might be the case that *"non-traditional"* is referring to the use of Cosmic Ray Neutron probes, but this is not obvious. In contrast, "human-practice" data is defined in Page 6, line 13 to be the irrigation amount. However, it is not clear if this term is referring to the GRIPC dataset used throughout the study (which is

based not only on human data, but also on remote sensing data), or to the amount of irrigation applied at two sites, as mentioned in Page 6, line 10.

We consider the 'human-practice' dataset to be the information on irrigation amounts and timing and the 'non-traditional' dataset to be the Cosmic Ray Neutron Probe datasets, both stationary and gridded. To clarify this, as well as to provide more details about the evaluation data in response to General Comment 3 below and several of Reviewer 4's comments, a new section has been added to the Methods called 3.2 Evaluation Data.

This new section begins (Page 8, Lines 19-20):

"The non-traditional, CRNP soil moisture data products and human-practice data gathered in Franz et al., (2015) are used to evaluate the sprinkler irrigation algorithm in LIS."

In this new section, with respect to the human-practice data and irrigation amount description (comment 3 below), the manuscript now reads (Page 8, Line 20-21):

"Human-practice data in the form of the irrigation amount and dates of irrigation application at one irrigated soybean and one irrigated maize site were reported via personal communication to Franz et al., (2015)."

Also with respect to the non-traditional dataset clarification, this section now reads (Page 9, Line 1):

"Additional non-traditional data from Franz et al., (2015) include a soil moisture product that uses the spatiotemporal statistics of the observed soil moisture fields…"

2. Related to the previous comment: although a reference is given for the GRIPC dataset, a brief description of this dataset would benefit the manuscript. An estimate of the uncertainties related to this dataset would also be helpful.

The following sentences describing the GRIPC have been added to Page 7 Lines 17-23:

"The GRIPC dataset integrates remote sensing, gridded climate datasets, and responses from national and sub-national surveys to estimate global irrigated area. The dataset closely agrees (96% at 500 m) with the USGS MIrAD-US2007 dataset (Pervez and Brown, 2010) and assessment of GRIPC against field level inventory data showed an 84% agreement in Nebraska (Salmon et al. 2015). This dataset represents a significant improvement in defining irrigated areas as compared to previous configurations of this model and scheme (Lawston et al. 2015) in which irrigated areas were defined using the 24-category USGS landcover classification, based on data from the 1990's. However, the GRIPC dataset overestimates irrigation intensity in the study area,…"

3. Also related to the first comment: a description of the irrigation data from the study in Franz et al. (2015) is also worth including. This is especially important in Figure 7, where irrigation at the maize site is shown, as well as in the text (Page 12, lines 14-16).

   *A description of the irrigation data has been included in the new 'Evaluation Data' section (3.2). Please see comment #1.*

4. The methodology for defining the growing season was not included in the Methods section. It is however mentioned later in the Discussion section on Page 15, lines 13-14 *"The method for determining the start and end of the growing season, based on the 40% annual range in climatological GVF, proved to be reliable for this study area and climate"*.

   *The details of the determination of the irrigation season have been added to the Methods section when first introduced. Page 10, Line 3 now reads:*

   > *"The growing season, addressed in question three, is a function of the gridcell GVF (i.e., 40% annual range in climatological GVF; Ozdogan et al. 2010)…"*

**Minor comments**

1. Page 2, line 9: (referring to observational data) *"are generally not obtainable at the scale of LSMS"* and Page 5, lines 19-21: *"available at the same spatial scale as LSMs"*

   What do the authors mean by scale of LSMs? land surface models can be run at a great range of scales. Perhaps the authors are talking specifically about high-resolution LSMs such as in this study? If so, please specify this.

   *Yes, we mean high-resolution but also are referring to the fact that observation data are often not available in spatially continuous/gridded fashion. This has been clarified:*

   > *Page 2, Line 9-10: "…are generally not obtainable in a spatially continuous format at the scale of high-resolution LSMs..."*

   > *Page 5, Line 23: "...area average soil water content…available at the same spatial scale as high-resolution LSMs"*

2. Page 4, line 1: *"For example, a flood irrigation parameterization. . . "*

   It is not clear if this is referring to scheme number 1 or 2 defined above in Page 3, lines 19-22. The text would benefit if this term ("flood irrigation parameterization") would be included in Page 3, lines 19-22 where applicable.

   **A sentence has been added to clarify here. The sentence at Page 3 Line 20 now reads:**

"This need has been addressed via irrigation parameterizations in LSMs that largely fall into three types of schemes: 1) defined increases to soil moisture in one or more soil layers (Kueppers and Snyder, 2011; de Vrese et al. 2016), sometimes referred to as flood (Evans and Zaitchik 2008),…"

3. Figure 1: The titles in each sub-figure are confusing. Perhaps the titles could read (top left, top right, bottom left, bottom right): "GRIPC irrigation intensity", "Tuned irrigation intensity", "Climatological GVF", and "Real-time GVF" to better identify what is being shown. Furthermore, the figure would improve by the inclusion of labels "a", "b", "c" and "d". Finally, the colorbar for the top figures (which is the same for both) could be shown in the center as it was done for the bottom figures.

All of the suggested changes have been made to Figure 1:

[Figure]

4. Page 11, line 2: *"the SPoRT GVF is greater than climatology in June"*
   Please clarify: do the authors mean "greater than climatological GVF"?

   Yes, this has been changed in the manuscript to 'greater than climatological GVF.'

5. Page 11, lines 3-4. *"However, in September, the SPoRT GVF detects the (negative) vegetation response to the July drought and irrigation amount and flux impacts are reduced"*.

What do you mean by "the sport GVF detects the negative vegetation response to the July drought?" is it because it is a real-time product as opposed to the climatological product and the fact that 2012 was particularly dry?

Yes, exactly. This has been rephrased to clarify:

> "…the SPoRT GVF detects vegetation stress caused by a July flash drought, resulting in reduced GVF, irrigation amounts, and flux changes."

6. Page 11, lines 4-7. *"These seasonal scale impacts illustrate that the NLDAS-2 forcing (e.g. precipitation) data, via changes to soil moisture, drives the irrigation timing during the growing season and that the behavior of the irrigation scheme is consistent with expectations of human triggering of irrigation during dry and wet periods".*
I am not sure I follow completely what is meant here. Is this saying that we expect irrigation triggering when there is no (or small amounts of) precipitation and no triggering when there is? If so, then this is already phrased better in the next lines (page 11, lines 9-10): *"At the interannual and seasonal scale, irrigation amounts and impacts are driven primarily by background rainfall regime, given by the forcing precipitation, with only small changes evident between the methods".*

Yes, the first sentence is meant to convey that irrigation is being triggered when there is little precipitation, as we would expect farmers to do. The second sentence is meant to re-iterate the triggering but also to point out that all three irrigation simulations had very similar results at the interannual and seasonal scales. This is set up as a contrast to the forthcoming daily scale results that show much larger differences in fluxes between irrigation experiments.

The first sentence has been re-worded to clarify:

> "These seasonal scale impacts illustrate that the NLDAS-2 forcing (i.e., precipitation) data, via changes to soil moisture, constrains the irrigation timing during the growing season, and that the soil moisture threshold is sufficient in triggering irrigation during rain-free periods"

7. In Figure 7, why not include the soil moisture from the CRNP gridded soil moisture product as well for comparison with the fixed probes?

We compared the CRNP gridded soil moisture time series to the CRNP stationary probes at the three sites and noticed that the gridded product had a small dry bias. This is confirmed by the Franz et al. (2015) paper that also notes a small dry bias in the gridded product that is likely a result of the rover driving on and sensing drier, gravel roads. This is in contrast to the CRNP stationary probes that are "painstakingly calibrated." Since the goal of this figure was to illustrate the impact of irrigation on the soil moisture time series and how well those changes are reproduced by the model, we show only the best available observations at these two sites, which are the CRNP stationary probes. The utility of the gridded product lies in the

areas where we don't have the probe data and as such, we use it to get a better understanding of how the model performs over the larger area (rather than at the individual sites).

8. Page 13, lines 11-12 *"In this study, we modify the spatial regression technique to treat irrigated and non-irrigated areas differently by using the CRNP (irrigated) rainfed data in the regression for (irrigated) non-irrigated gridcells"*.
I am not sure I follow the last part with the parentheses "by using the CRNP (irrigated) rainfed data in the regresion for (irrigated) non-irrigated gridcells". Could you please clarify this?

Please see comment #9

9. Referring to the same text in the last comment, in my opinion, since this is also a novel contribution (the modification of the spatial regression technique for the gridded product), a comparison between the previous and the new product could be included as supplementary material.

In response to both comments 8 and 9, this section has been rephrased, expanded upon, and relocated to Page 8 Line 19- Page Line 13 in the new Section 3.2 (Evaluation Data). It now reads as follows:

"Additional data from Franz et al., (2015) include a gridded soil moisture product that uses the spatiotemporal statistics of the observed soil moisture fields, as obtained via the CRNP rover surveys, and a spatial regression technique to create a 1-km, 8-hour gridded soil moisture product for the growing season (May – Sept, 388 values). Franz et al., (2015) used the average of the three stationary CRNP probes as the regression coefficient, which can smear the spatial differences between irrigated and rainfed areas. In this study, we modified the spatial regression technique to treat irrigated and non-irrigated areas differently by using the CRNP rainfed probe in the regression for non-irrigated gridcells and the average of the two irrigated CRNP probes for the irrigated gridcells. This results in a gridded soil moisture product that retains the spatiotemporal differences of the rainfed and irrigated areas. Irrigated and non-irrigated gridcells are defined by an estimated irrigation mask created using the landcover map of Franz et al. 2015 from ground observations. A comparison of the original and new regression products at an irrigation and non-irrigated point is given in the Supplement.

As the text states, the following figures have been added to the supplement to show the difference between the new and original regression products. With the original regression technique (a) few differences are seen between the irrigated and rainfed points, especially during the dry-down period in late July to early August. The averaging of the probes results in a levelling off of soil moisture during this time. (b) The new regression technique results in the non-irrigated point showing decreasing SWC during the dry down period, as at the CRNP

rainfed probe, while the irrigated point shows increasing SWC due to irrigation during the dry down. This explanation has been added to the supplement figure caption (below).

[Figure]

**Supplement 1.** Time series of soil water content at an irrigated and non-irrigated point given by the gridded CRNP product using (a) the original regression from Franz et al., 2015 (b) the new regression used in this study that treats irrigated and non-irrigated areas differently. With the original regression technique (a) few differences are seen between the irrigated and rainfed points, especially during the dry-down period in late July to early August. The averaging of the probes results in a levelling off of soil moisture during this time. (b) The new regression technique results in the non-irrigated point showing decreasing SWC during the dry down period, as at the CRNP rainfed probe, while the irrigated point shows increasing SWC due to irrigation during the dry down.

10. Page 13, lines 17: *"during which irrigation was applied at the irrigated maize site"*

Only at the maize site? or the whole domain shown in Figure 1? The caption reads *"when irrigation was applied at the irrigated maize and soybean sites"*. To my under- standing, the maize site and soybean sites are only parts of the whole domain, and this figure (Figure 8) is showing a spatial comparison of the whole domain.

Yes, the reviewer is correct. The figure showed the whole domain, which includes the irrigated sites, but is not exclusively the irrigated sites. The intention for that statement was to emphasize that the CRNP gridded observations are at least partially impacted by the irrigation that we know is occurring in at least some areas on that day. This figure has been changed from a CDF to a scatterplot as per Reviewer 3's comments and the caption has been reworded to that below with the reviewer's comments in mind:

"Figure 8. Scatterplot of the gridcell soil moisture content (volumetric) given by the irrigation simulations as compared to the CRNP gridded soil moisture product."

[Figure]

11. Figure 8: In the legend, consider changing "CoSMOS" to "CNRP" to be consistent with the rest of the paper.

The legend has been updated in the new Figure 8. Please see previous comment (#10).

12. Page 14, lines 9-10 *"Furthermore, when irrigated and non-irrigated areas are averaged separately, the irrigated (Control) simulations match the distribution of irrigated (non-irrigated) areas well"*.
Again, I do not understand the use of the parenthesis here "irrigated (non-irrigated)".

This sentence has been rephrased:

> "Furthermore, when irrigated and non-irrigated areas are averaged separately, the irrigated and control simulations match well the distribution of irrigated and non-irrigated areas, respectively (Fig. 9b)"

**4 Technical corrections**

All of the following technical corrections have been made.

1. Page 7, line 19 *"as evidenced by only 5% of the gridcells having intensity less than 100% (Fig 1)"*
I think this should be "Fig 1a" instead of "Fig 1".
2. Page 7, line 22 *"(i.e. observationally tuned: Fig 1)"*
I think this should be "Fig 1b" instead of "Fig 1".
3. Page 7, lines 19 and 22 and Page 8, line 4.
Check consistent use of either "(Fig X)" or "(Figure X)".
4. Figure 6: Label in Y-axis "Change in Domain Avg Qle" instead of "Doman"
5. There is a dot missing in Page 13, line 24 before "The model distributions do not match the CRNP CDF, which instead shows. . . "

---

## Author Comment (AC2) · 20 Apr 2017

We would like to thank all of the reviewers for the thorough and insightful suggestions and comments. We made substantial changes to the manuscript, replaced one figure, and completed an additional model simulation in response to the feedback we received. We feel that the manuscript has improved significantly as a result of these thoughtful reviews. Please find our detailed responses to the reviewer's comments below.

Please note that the **reviewer comments are shown in black** and our author responses are in blue. Where changes have been made in the manuscript, the page and line number(s) are given. In some cases, to highlight changes to passages in the manuscript, these sections are copied and pasted from the manuscript.
* * *
**Reviewer #2:**

**Summary:**

The authors provide a useful and clearly-written evaluation of irrigation simulated by an advanced Land Surface Model. These types of evaluation are in short supply, and the use of CRNP in model evaluation is, to my knowledge, novel and potentially quite useful. I believe that the Discussion Paper is of sufficient interest and quality for publication in HESS. That said, the numerical experiments presented in the study are rather limited. Sensitivity to GVF dataset and irrigation intensity factor are evaluated, but none of the many other factors that the authors list are explored. This may lead to the wrong impression that the tested factors are the most important when simulating irrigation, when I see no evidence presented by the authors that this is in fact the case. Ideally, the authors should present a more inclusive set of sensitivity tests to inform future modeling studies about the relative importance of different factors. If this is not possible, or if the authors view it as unnecessary, then a more convincing justification for the choice of experiments is required.

**General Comments:**

1. Meteorological Forcing: In the abstract and at several other passages in the text the authors emphasize the importance of high quality meteorological forcing data for accurate simulation of irrigation. Their results suggest that NLDAS is high quality, as shown most convincingly by the temporal match of simulated irrigation to spikes in observed soil moisture. I believe that NLDAS is high quality and that these results show impressive performance at local scale. But I'm not sure that the authors can actually make any conclusions about the importance of forcing data to irrigation simulations, given that they do not compare NLDAS simulations to simulations with any lower quality forcing dataset. Yes, it its intuitive that simulations with NLDAS will be better, but the numerical experiments don't demonstrate this, and they don't show us *how* important it is. This is particularly the case when one considers spatial or temporal scale. The authors nicely demonstrate that simulations are more realistic at larger and longer scales than they are at local and shorter scales. How important is meteorological forcing if we are concerned with large and long time scales? Additional simulations with an alternative, poorer quality meteorological

forcing dataset would be the obvious way to test this, but the authors might find other ways to make the point.

The foundational study for this work, Ozdogan et al., (2010), evaluated this scheme at larger (continental U.S.) and longer (yearly) time scales with annual water withdrawals and county level data. For this study, the primary interest is in evaluating the scheme performance at smaller and shorter timescales, so a robust evaluation of the meteorological forcing at large and long timescales is beyond the scope of this work. With respect to the support for the NLDAS2 conclusions, however, the reviewer raises some good and justified questions.

In response, we have completed an additional run equivalent to the Standard irrigation run in all aspects (e.g., GRIPC irrigation intensity, climatological GVF) except that we used GDAS meteorological forcing instead of NLDAS2. GDAS is coarser resolution (1/4 degree) and does not include rain-gauge corrections. GDAS supplies a greater total amount of precipitation in the May through July time period. See figures:

[Figure]

[Figure]

The greater total amount of precipitation from GDAS results in a wetter soil column leading up to and throughout the mid-to-late July rain-free period, delaying the onset of irrigation triggering by the scheme. As a result, the soil moisture starts out wetter in mid-July than the other irrigation simulations (forced with NLDAS2) and even the CRNP, then dries out to a level below that of the other schemes (as a result of moisture being sustained in the root zone and prohibiting irrigation). The irrigation is finally triggered at the beginning of August, a few days prior to the return of precipitation to the area. See figure below (top layer soil moisture):

[Figure]

This simulation adds support to the conclusion that accurate precipitation data is essential to constrain the irrigation triggering. A brief description of this additional run has been added to the discussion section.

The newly added part of the Discussion (Page 15, Line 19 – Page 16 Lines 1-5) reads:

"For this small domain, the NLDAS2 precipitation proved to be sufficiently accurate, matching well that given by the nearby York, Nebraska AWDN. However, for other regions, reliable meteorological forcing may not be available. To further explore the impact of the forcing precipitation on the irrigation triggering, an additional simulation was completed that is equivalent to the Standard irrigation run in all aspects (e.g., GRIPC irrigation intensity, climatological GVF) except that the Global Data Assimilation System (GDAS) meteorological forcing is used rather than NLDAS2. In contrast to NLDAS2, GDAS is coarser resolution (1/4 degree) and does not include rain-gauge corrections. Results show that GDAS supplied a greater amount of total of precipitation in May through July 2014, creating a wetter soil column and prohibiting irrigation triggering in mid-to-late July, in contrast to observations and the other irrigation simulations. As a result, the soil moisture dynamics of the GDAS simulation at the maize site differ substantially from the CRNP observations and the NLDAS2-forced simulations. These results underscore the

need for highest quality datasets available for the area of interest, which for this region and time frame was NLDAS2."

2. Thresholds: The authors appropriately emphasize the importance of selecting proper thresholds for soil moisture and GVF at several points in the text. But the manuscript does not offer any evaluation of either. In both cases a single threshold is applied and attributed to previous studies. It would be quite interesting to know how the impact of using different GVF datasets compares to differences caused by small changes in GVF threshold. And how does a modest change in threshold impact total water use, as compared to the tested sensitivity to prescribed irrigation intensity?

The sensitivity of the irrigation scheme to the soil moisture and GVF thresholds has already been examined in the Ozdogan et al., (2010) for a larger area that includes our study region. The 50% of field capacity soil moisture triggering threshold was selected by their study as being most appropriate based on discussions with local experts, including some in Nebraska, as well as through trial and error (Ozdogan et al., 2010). As this is the same scheme used here, we didn't consider it necessary to re-test the SM threshold and instead accepted it as being the best for this region based on current literature. The accurate timing of irrigation triggering shown in the results supports that this threshold was reasonable.

Although the gridcell GVF *value* is used to calculate the crop root zone and to scale the amount of water applied, the GVF *threshold* is only used to determine the start and end of the irrigation season. As a result, a small change in the GVF *threshold* would only increase or decrease slightly the length of the irrigation season. The GVF threshold for our region gives an appropriate irrigation season of June – September, so we didn't consider it necessary to change this threshold at all.

I understand that no study can be comprehensive on all parameters, but I don't fully understand why the authors chose to look only at GVF dataset in GRIPC irrigation intensity when other subjective modeling decisions might have as large or larger impacts on the simulations. If possible I would encourage the authors to expand their sensitivity test in order to justify the selection of these two factors as the focus of study.

The main objectives of this study were not necessarily to turn every knob, but instead to take the best available collection of default datasets we have (e.g., those that someone new to model would probably choose) and to see how well it performs (i.e., the Standard run). Then secondarily, to determine if it is possible to improve upon that standard model performance by either 1) incorporating additional information to tailor the datasets to our study area (Tuned irrigation intensity), or 2) by using a new and improved GVF dataset (SPoRT) that detects vegetation response to soil stress. Rather than a blanket sensitivity study, these were targeted in areas where we knew we could improve the model/datasets based on solid information.

The focus on irrigation intensity and GVF datasets for potential improvement to model performance is two-fold:

1) Irrigation intensity and GVF are critical to **both** the *triggering* of irrigation and the calculation of the *amount* of irrigation water applied. As a result, flaws in the scheme could be made more apparent by switching out these datasets. Additionally, these two datasets (SPoRT GVF and GRIPC irrigation intensity) are brand new and have not been used with an irrigation scheme until now.

2) The other datasets that play a role in irrigation triggering, (i.e., landcover, soil texture, soil type, crop type) were by default homogeneous across the study area and were appropriate for the area based on the ground truth we had. For example, the landcover for every grid cell in the domain was 'croplands'. At 1 km resolution, there is not a better classification of these gridcells than cropland (e.g., even the gridcells that contain small buildings or roads still occupy < 50%; croplands is dominant land use). Similarly, we didn't have additional information to be able to improve upon the default soil type or texture. With regards to crop type, the data from Franz et al., (2015) showed 81% maize and 19% soybean, in contrast to 100% maize in the default crop type map. As a result, we did an additional run with tuned crop type and altered max root depth. The results of this run are presented as a note in the Discussion (Page 16, Lines 12-21) rather than featured prominently. This was done with the intention of simplicity (i.e., to minimize confusion that could be caused by introducing another iteration) because this run was not significantly different than the other irrigation runs.

**Minor Comments:**

Page 3, line 20: This list of options misses flood irrigation simulation (unless it's supposed to be covered by #1). Several studies have employed flood irrigation, including Yilmaz et al. (2014), Leng, and Evans & Zaitchik (2008).

The intent was for flood to be covered by #1. The sentence has been edited to clarify:

Page 3, Line 20: "1) defined increases to soil moisture in one or more soil layers (Kueppers and Snyder, 2011; de Vrese et al. 2016)., sometimes referred to as flood (Evans and Zaitchik, 2008),"

Section 2.3: It would be useful to include a sentence or two on why CRNP measure- ments are sensitive to soil moisture. Many readers (myself included) are not deeply familiar with this technique.

An additional sentence has been added to section 2.3 (Page 5 Lines 18-20) addressing this point (bolded):

"The theoretical basis for the CRNP method follows that fast neutrons injected into the soil by the CRNP will be slowed more effectively by collisions with hydrogen atoms present in soil water than by collisions with any other element (Visvalingam and Tandy,

1972). **Thus,** the neutron density measured by the probe is inversely correlated with soil moisture…"

---

## Author Comment (AC3) · 20 Apr 2017

We would like to thank all of the reviewers for the thorough and insightful suggestions and comments. We made substantial changes to the manuscript, replaced one figure, and completed an additional model simulation in response to the feedback we received. We feel that the manuscript has improved significantly as a result of these thoughtful reviews. Please find our detailed responses to the reviewer's comments below.

Please note that the **reviewer comments are shown in black** and our author responses are in blue. Where changes have been made in the manuscript, the page and line number(s) are given. In some cases, to highlight changes to passages in the manuscript, these sections are copied and pasted from the manuscript.
* * *
**Reviewer #3:**

**I. Summary**

This manuscript examines the issue of developing and validating realistic irrigation schemes for use in land surface models (LSMs). In this study, the authors utilize observation-based datasets of irrigation intensity and green vegetation fraction (GVF) to tune the LSM irrigation amounts, which are validated against data obtained from Cosmic Ray Neutron Probes (CRNP). The main conclusion of the authors is that the timing, amount, and spatial spread of irrigation are more sensitive to the choice of irrigation scheme at smaller spatiotemporal scales than at larger, more typical scales for regional climate models. Given the balance of evidence presented and the use of a novel dataset (CRNP) for addressing this issue, it seems that the authors have arrived at robust and meaningful conclusions that would be worthwhile additions to HESS and to the field of hydrology, in general. While I have no major qualms with the content or substance of the manuscript, I do present below some more minor comments for improving the robustness and presentation of the results.

**II. General comments**

A. NLDAS-2 – The authors mention several times throughout the manuscript the need for "high-quality" meteorological forcing and point out repeatedly the accuracy of the precipitation data from NLDAS-2 for their domain. While it certainly seems that NLDAS-2 provides accurate forcing over this domain (and is a high-quality dataset, in general), I echo Reviewer 2 in cautioning against drawing far-reaching conclusions about NLDAS-2 from this limited study. The entire study domain is 15 x 15 km, very small even for typical regional climate model simulations; the entire domain would fit in 4 grid cells of NLDAS-2 (1/8 degree horizontal resolution). Is there evidence that NLDAS-2 would provide equally accurate data for a different domain within the same region, or in a different region or year? If so, then I would provide a sentence or two explaining the skill of NLDAS-2 over the general region (e.g., Great Plains/Midwest) during the growing season (perhaps from the Xia study). If not, then please temper the language emphasizing the high quality of NLDAS-2 with the understanding that the spatial domain of this

study is extremely limited and that NLDAS-2 may not be as accurate in other agricultural regions in North America.

As per reviewer 2's suggestion, an additional run with GDAS forcing was completed and a brief description of the results is now included in the Discussion section (Page 15, Lines 9-15 to Page 16 Lines 1-5). In this newly added paragraph, we've taken care to add qualifiers to the NLDAS2 statements. The section now reads as follows, with bolded words to emphasize the tempered language about NLDAS2:

> "**For this small domain**, the NLDAS2 precipitation proved to be sufficiently accurate, matching well that given by the nearby York, Nebraska AWDN. **However, for other regions reliable meteorological forcing may not be available.** To further explore the impact of the forcing precipitation on the irrigation triggering, an additional simulation was completed that is equivalent to the Standard irrigation run in all aspects (e.g., GRIPC irrigation intensity, climatological GVF) except that the Global Data Assimilation System (GDAS) meteorological forcing is used rather than NLDAS2. In contrast to NLDAS2, GDAS is coarser resolution (1/4 degree) and does not include rain-gauge corrections. Results show that GDAS supplied a greater amount of total of precipitation in May through July 2014, creating a wetter soil column and prohibiting irrigation triggering in mid-to-late July, in contrast to observations and the other irrigation simulations. As a result, the soil moisture dynamics of the GDAS simulation at the maize site differ substantially from the CRNP observations and the NLDAS2-forced simulations. **These results underscore the need for highest quality datasets available for the area of interest, which for this region and time frame was NLDAS2.**"

**III. Specific comments**

A. Page 14, line 10 – "These results suggest that if this domain were one gridcell in a larger, coarser resolution domain (e.g. 15 km spatial resolution), the variation in the gridcell soil moisture (given here by the domain average) over the growing season would be representative of observations."

It would be interesting to see a supplemental model analysis with coarser-resolution grid cells (either in this paper or a future one) that validates this hypothesis. For example, what is the spatial threshold at which large-scale forcings begin to dominate the changes in the soil moisture signal?

We agree that this is an interesting question and appreciate the suggestion! This will certainly be an area of future study using the flexibility of the LIS system (resolution, forcing, and inputs).

B. Page 15, line 9 – ". . .indicating that the model is quite insensitive to the maximum root depth change. . ."

Some common irrigated crops, such as alfalfa, have max root depths of 2+ meters. Though irrigated alfalfa is much less common in Nebraska when compared to corn and soybeans, it would be instructive to not make the above claim about the insensitivity of the model to max root depths unless other crops with much larger or smaller max root depths have been tested.

This sentence has been rephrased to emphasize that the root depth sensitivity tested was only for a small change to a specific crop:

> "The results of this analysis showed little difference between this simulation and the others, indicating that the model is insensitive to small changes (up to 20%) in the maximum root depth. However, land surface models that have a more complex treatment of crops, study areas with greater heterogeneity of crop types, or experiments that replace a particular crop with one that has a vastly deeper root system, are examples beyond the scope of this study that could potentially result in a greater sensitivity of the model results to crop root depth."

C. Page 15, line 22 – ". . .a growing number of options for irrigation intensity datasets in the coming years".

A new global irrigation dataset (the Historical Irrigation Dataset) was published through HESS rather recently (S. Siebert, M. Kummu, M. Porkka, P. Döll, N. Ramankutty, and B. R. Scanlon (2015), "A global dataset of the extent of irrigated land from 1900 to 2005," Hydrology and Earth System Sciences. DOI: 10.5194/hess-19-1521-2015). It may deserve a citation here because of its recent development and global coverage.

It is certainly appropriate and has been added.

D. Figure 1 – Are the spotty areas of low irrigation intensity in the Tuned plot over urban areas? A brief explanation of this in the text may be warranted.

The spotty areas indicate the irrigation intensity has been reduced due to the presence of roads, wetlands, rainfed fields, and/or buildings. Of the three gridcells with 0% irrigation intensity, two contained mixed-use land, small buildings, and roads (though, not built up enough to really be considered 'urban'). The remaining 0% irrigation intensity gridcell contains the rainfed site given in the CRNP observations.

The figure caption has been updated as follows:

> "**Figure 1.** (a) GRIPC irrigation intensity (percent) given by Salmon et al. (2015) used in the Standard and SPoRT simulations and (b) the observationally tuned irrigation intensity used in the Tuned simulation. **The spotty nature of Tuned indicates irrigation intensity has been reduced due to the presence of roads, wetlands, rainfed fields, and/or buildings**. Also shown is the average greenness vegetation fraction (unitless) in July 2012

given by (c) NCEP climatology used in the Standard and Tuned simulations and (d) SPoRT real-time dataset used in the SPoRT run."

E. Figure 2 – It would be helpful to mention in the figure caption that SPoRT uses the climatological GVF in years 2009 and 2010 (as is already mentioned in the text) to avoid confusion.

The caption has been updated to include this information (bolded) and now reads:

> **Figure 2.** Domain and monthly averaged GVF from the NCEP climatological GVF dataset, used in the Standard run, the SPoRT GVF dataset used in the SPoRT run, and the difference between the two (SPoRT – Climatology). **As the SPoRT dataset is not available prior to 2010, the long-term SPoRT simulation uses climatological GVF for 2009-2010, and the SPoRT GVF dataset is incorporated in December 2010 and used throughout the rest of the simulation."**

F. Figure 4 – I don't believe that IRR was ever defined (in either the main text or the figure caption).

For all captions, "IRR - Ctrl" has been replaced with (i.e., each irrigation run minus Controll). The legend have been updated in Figures 7 and 9 so that they don't include IRR and are consistent with the other legend labels.

G. Figure 4 – The boundaries of Layer 4's soil depths are only mentioned here, not in the main text. Since crop roots barely extend into this layer (max root depths of 1 or 1.2 m), perhaps this further explains why there seems to be much more variability in soil moisture between irrigation simulations in Layer 3 than in Layer 4.

Yes, the reviewer is correct. To call attention this fact, Page 11, Lines 8-9 have been updated to read:

> "Increases in the third soil layer, **which includes the root zone**, are quite consistent annually with a near doubling of the soil moisture when irrigation is turned on."

H. Figure 8 – I think that the presentation of "spatial" CDFs in this figure is rather non- intuitive. To me, it would be much more intuitive to see the differences in the spatial distributions of soil moisture within the domain using a histogram, especially since each CDF is plotted for only a single time step and thus there is no "accumulation" of data over time. In this figure, since data is accumulated spatially (in two dimensions) rather than temporally (in one dimension), the shape of the CDF would be rather arbitrary and would partly depend on the order in which you spatially sample the grid cells.

Thanks much for this suggestion. This figure has been changed to a scatterplot of the gridded observations versus the LIS simulations:

[Figure]

The text has been updated accordingly and all mentions of 'temporal CDF' have been changed to 'CDF':

"The LIS-simulated soil moisture variability in time and space is evaluated against the CRNP gridded soil moisture product, described in Sect. 3.2. The spatial variability is assessed first with a scatterplot generated using all gridcell soil moisture values from the LIS simulations and the modified CRNP product aggregated at 4, 12, and 20 UTC on 25 July 2014 (Fig. 8). Next, the temporal variability is assessed using a CDF of the domain-averaged soil moisture values from May 5 to Sept 22 at 8-hour intervals (Fig. 9).

Figure 8 shows that the Control simulation does not match the observations in magnitude or variability, instead showing uniformly dry soil across the domain (e.g., range of 0.01 versus more than 0.1 in observations). The spatial variability is increased in the irrigated simulations, but these runs exhibit jumps between clusters of values as a result of irrigation triggering and dry down across the domain. The different levels of clustering shown by the irrigated simulations are a result of the input parameter datasets, as triggering and timing are dependent on these datasets. Although the Control

simulation is too dry, the irrigation overcompensates and increases the soil moisture to levels beyond that shown in the gridded observations. These results suggest that the model, even with the irrigation algorithm turned on, is not able to accurately simulate the small-scale (i.e., field scale) heterogeneity in soil moisture that is present in the CRNP data…"

I. Figure 8 – Neither the figure nor the figure caption explain what is being plotted in the figure. Units would also be appreciated (even if unitless).

Please see notes above notes about the updated Figure 8.

**IV. Technical corrections**

**All of the following technical corrections have been made as suggested. We thank the reviewer for his/her attention to detail.**

**With respect to comment "S" below, the text has been changed to:**
"In this study, we modified the spatial regression technique to treat irrigated and non-irrigated areas differently by using the CRNP rainfed values in the regression for non-irrigated gridcells and the average of the irrigated CRNP probes for the irrigated gridcells.

A. Page 1, line 17 – "at the interannual scale, but become. . ." – Remove the comma.
B. Page 2, line 23 – "previous evaluation efforts, and introduces…" – Remove the comma.
C. Page 3, line 14 – e.g., "de Vrese et al. 2016" – Please be consistent with placing commas after "et al." in internal citations.
D. Page 4, line 1 – "with a two different. . ." – Remove "a".
E. Page 4, line 2 – "in the U.S. Central Great Plains. . ." – "Central" should be lowercase.
F. Page 4, line 5 – "Tuinenburg et al., 2014), or in. . ." – Remove the comma after "2014)"
G. Page 4, line 15 – No need for commas surrounding "such as these".
H. Page 4, line 23 – ". . .to reproduce county and water resource region irrigation water usage. . ." – Change to ". . .to reproduce irrigation water usage within counties and water resource regions. . .".
I. Page 5, line 17 – Change "c.f." to "cf.".
J. Page 5, line 19 – "reliable, area-average soil water content" – Throughout the manuscript, please change to "area-averaged" or "domain-averaged" (as in the above example) when being used as an adjective and "area average" and "domain average" when being used as a noun.
K. Page 6, line 9 – Change to "Sect. 3".
L. Page 7, line 22 – "i.e. observationally tuned" – Change all instances of "i.e." and "e.g." to "i.e.," and "e.g.,".
M. Page 8, line 8 – "more sophisticated, but computationally expensive. . ." – Remove the comma.
N. Page 8, line 8 – "such a dynamic. . ." – Change to "such as".
O. Page 8, line 14 – Change to "bias-corrected".

P. Page 11, line 14 – "the SPoRT run increases latent heat flux by more than 100 W mˆ-2 more than Standard" – Change to "latent heat flux in the SPoRT run is more than 100 W mˆ-2 greater than Standard".

Q. Page 12, line 15 – Add a space between "mm dayˆ-1" and "(not shown)".

R. Page 12, line 25 – Add a comma after "(e.g., satellite)".

S. Page 13, line 11 – "CRNP (irrigated) rainfed data. . ." – I would discourage this par- enthetical style (it already seems to have confused other reviewers). If you must use it, I would recommend putting the parenthetical expression second, e.g., "CRNP irrigated (rainfed) data". However, I would instead prefer this and related sentences to be written as: "by using the CRNP irrigated and rainfed data in the regression for irrigated and non-irrigated gridcells, respectively".

T. Page 13, line 23 – Add a period after "dependent on these datasets".

U. Page 13, line 25 – Change "exhibit" to "exhibits".

V. Page 14, line 5 – Hyphenate "deficit based".

W. Page 14, line 11 – Hyphenate "coarser-resolution".

X. Page 16, line 3 – Remove the comma after "LSM framework".

Y. Page 16, line 4 - Remove the comma after "latent heat flux".

Z. Page 16, line 21 – Remove the comma after "soil moisture".

AA. Page 16, line 23 – Change to "USDA Census of Agriculture".

BB. Page 17, line 1 – Hyphenate "satellite based".

CC. Page 17, line 2 – Add period after "(Kumar et al., 2015)".

DD. Page 17, line 4 – Change "premiere" to "premier".

EE. Page 17, line 23 – Capitalize "a" after Myhre, and ditto for all other instances of mixed case for author names in the reference list.

FF. Page 18, line 8 – Be consistent with italicizations: Either italicize all journal names or keep them all as plain text.

GG. Page 18, line 8 – Change "hess" to "HESS".

HH. Page 18, line 28 – What does "Received" mean?

II. Page 19, line 3 – Be consistent with capitalization of the article titles: Either capitalize only the first word and proper nouns (standard practice) in every title or capitalize all words in every title.

JJ. Page 20, lines 5-9 – I think that these lines are in a slightly different font than the other references.

KK. Page 21, lines 2-3 – See above comment.

LL. Page 21, line 23 – What is "Artn"? Article number?

MM. Fig. 1 caption – Please define the units of irrigation intensity (even if unitless).

NN. Fig. 4 caption – Add a colon after "LSM default layers".

OO. Fig. 4 caption – Be consistent with parenthetical notes: Delta Z is included for the middle layers but not for the top or bottom layers.

---

## Author Comment (AC4) · 20 Apr 2017

We would like to thank all of the reviewers for the thorough and insightful suggestions and comments. We made substantial changes to the manuscript, replaced one figure, and completed an additional model simulation in response to the feedback we received. We feel that the manuscript has improved significantly as a result of these thoughtful reviews. Please find our detailed responses to the reviewer's comments below.

Please note that the **reviewer comments are shown in black** and our author responses are in blue. Where changes have been made in the manuscript, the page and line number(s) are given. In some cases, to highlight changes to passages in the manuscript, these sections are copied and pasted from the manuscript.
* * *
**Reviewer #4:**

**General comments:**

This is an interesting paper on the evaluation of an irrigation scheme within a land surface modeling framework. This is an area that needs research and I see this a potentially valuable contribution on the matter.

While generally well written, the structure and organization of the Background (particularly Section 2.3) and the Methods sections needs to be improved to ensure a better flow and enhanced readability. The study region, models, input datasets and evaluations should be described in a more logical and orderly manner with less intermixing. These issues are described in more detail in the specific comments below. The discussion section is very short and would benefit from more elaboration and high quality insights on the limitations and challenges as well as opportunities for irrigation modeling.

A new section has been added to the methods and more information has been added to the Discussion. Please see specific comments below.

Some of the used input datasets need more justification. GVF is an important dataset for the irrigation modeling but is reported at coarse resolution (3 and 16 km) inconsistent with the resolution of the LSM (1 km). Not clear to me why a 1 km based version isn't used here. The MODIS phenology product (produced at 500 m resolution) would probably be more useful in this context for establishing the start and duration of the growing season.

Please see specific comments 10 and 12 for detailed responses.

I'm also a bit concerned that 1 km isn't the most appropriate scale to do irrigation modeling and accuracy assessments as you will inevitable run into mixing of rainfed and irrigated fields given the characteristic size of the fields. LSM runs at 500 m resolution would probably have been

more appropriate, also considering the scale of the CRNP validation dataset, and feasible using widely available surface inputs generated at consistent resolutions.

Mixing of rainfed and irrigated fields is certainly an issue that arises in irrigation modeling, even at 1 km, which is considered high resolution for land-atmosphere interactions and regional weather modeling studies. However, 1 km is the highest resolution we can run, while still being appropriate and relevant to our broader goals.

The spatial resolution of 1 km is the most appropriate scale for this study for two main reasons:

1) The highest resolution input datasets we have are 1 km, so running at 500 m would not improve our results in this study; it would simply give the same information broken up into more gridcells.
2) The broader context goal of evaluating this irrigation scheme is for its later use in land-atmosphere interaction studies (Page 2, paragraph 1; Section 2.1). It is difficult and typically not advisable to run a coupled atmospheric model at 500 m, especially for land-atmosphere interaction studies. The behavior of the planetary boundary layer (PBL) in atmosphere/mesoscale models, such as WRF, is determined by the PBL parameterization. These parameterizations are not recommended for use at 500 m as some of their assumptions break down at such fine scales.

**Specific comments:**

1) Page 1 L14: Please define the scale associated with "high resolution"

High resolution is 1 km in this case.

2) Page 1 L19: What precisely does the "human practice data" consist of?

Human practice data is the irrigation timing and amount. This has been clarified in the newly added section 3.2 – Evaluation Data.

3) Page 1 L21: "two irrigated fields" – what irrigated fields are you referring to here (soybean and maize)?

Yes, this is clarified in the newly added section on Evaluation Data (3.2).

4) Page 2 L21 and L25: Please define what you mean by coarse and high resolution here.

5) Page 6 L1-7: This paragraph reads a bit confusing with mentioning of all the different temporal and spatial resolutions. A bit unclear what product version is used for the evaluation. Does the 12x12 km survey area correspond to the 15x15 km domain of this study? Why the domain difference?

All of the evaluation data is explicitly defined in the newly added Evaluation Data section (3.2). The 12x12 km survey area is contained entirely within the 15 x 15 km domain area of the study. The grid projection (UTM) used in the Franz et al. (2015) study is not directly compatible with the grid definitions in LIS. Therefore, since we couldn't recreate the exact grid, we made a slightly larger domain to ensure that the entirety of the Franz domain was contained within the LIS simulation domain.

6) Page 6 L8-16: This Section adds to the confusion by repeating some of the statements above and also adding additional evaluation datasets (human practice data etc.) not related to the CRNP (although that is the title of the Section). Differences between the CRNP and COSMOS datasets should be clarified, if any. The finishing paragraph relates the overall objectives and novelty of the work, which don't belong here. This Section requires some revision – the evaluation components might be more appropriately positioned in the method section. You may need a completely separate section for describing the additional datasets mentioned here.

COSMOS is the observing network of stations and rovers, while CRNP refers to the observing instrument. The first sentence of Section 2.3 makes the distinction:

> "A potential solution to fill the gap between point and remote sensing observations of soil moisture is the Cosmic-Ray Neutron Probe (CRNP) **method**, organized through the Cosmic Ray Soil Moisture Observing **System** (COSMOS), which **has ~200 probes operating** globally since 2011."

The source of the human-practice data is Franz et al. 2015, which is described in this section, and thus why the human-practice data is mentioned here.

We have shortened this paragraph by removing details of the evaluation data and have instead incorporated these details into a new section 3.2, called Evaluation Data, as suggested by the reviewer. The novelty statement has been moved to the last paragraph of the introduction.

7) Section 2.3: The CRNP data description is currently part of the introduction/background part of the manuscript. While it makes sense to mention and introduce the data as a useful validation source in this context, I feel that the detailed description of the actual dataset used here for evaluation purposes should be moved to a separate section in the Methods section (or Methods and data section). Here you could appropriately describe all the datasets used in the study.

Details about the CRNP data from the Franz et al. study used for evaluation in our study have been moved to a new sub-section of the Methods, called Evaluation Data (3.2), as per the reviewer's suggestion.

8) Section 3: I would start this with a description of the study area and domain to set the stage.
9) Section 3.1: I find this section quite confusing to read as it includes both modeling and

evaluation details and references to elements described in Section 3.2. I think you need to rethink the organization of the Method section adopting a more logical organization for improved flow and readability. Personally, I would prefer to have all model descriptions first before the description of experiments and evaluations to be performed.

This section does not include any evaluation details. It describes the land surface model and modeling framework (paragraph 1), the time period for the simulations (paragraph 2), introduces the four simulation experiments (paragraph 3), and then details the important distinctions between the four simulation experiments (remaining paragraphs).

10) Page 8 L1-5: So why isn't the GVF datasets provided at 1 km to be consistent with the LSM resolution?

The resolution of the NCEP climatological GVF used in this study is 1 km. The statement about the 16 km GVF dataset was included as part of the summary of results from Case et al., (2014); their study used 16 km climatological GVF. Admittedly, it did read like the climatological GVF used in this study is also 16 km. We removed these extra details from the Case et al., 2014 study description as they are unnecessary and added confusion. We also added the resolutions of the GVF datasets when introducing them. Page 8 Lines 2-5 now read (bold is newly added):

> "The SPoRT run makes use of the GRIPC irrigation intensity dataset, like the Standard run, but uses a real-time GVF product at **3 km spatial resolution** from NASA-MSFC's Short Term Prediction, Research, and Transition Center (SPoRT; Case et al., 2014). This is in contrast to the other runs that use climatological GVF at **1 km** from the National Centers for Environmental Prediction (NCEP)."

With respect to the resolution of input datasets more generally, we always use the best available, most appropriate input datasets for our application. Although we like to use high-resolution whenever possible, the highest resolution is not always the best available. This is the situation with our SPoRT dataset. Although the SPoRT GVF dataset is produced at 0.01 degree (~ 1 km), there was a change in the Continental US grid in Feb 2012 that impacted the 1 km dataset. We used the 3 km dataset instead of 1 km to avoid potential inconsistencies resulting from the grid change in 2012 (in the middle of our long-term spinup).

You also need to specify precisely what the GVF product is used for, when first introduced. From what I can read later in the manuscript it is predominantly used to determine the start and end of the growing season; couldn't you use the MODIS phenology product (see comment 12) more appropriately for this purpose? In addition, this product is available for the full duration of the study.

The GVF dataset is used in irrigation scheme in two main ways:

1) It is involved in the determination of the irrigation season, as the reviewer notes. This is a central feature of the Ozdogan et al. (2010) irrigation algorithm. While it is certainly

possible to use a different method, such as the MODIS phenology for determining the irrigation season, this would be a considerable deviation from the irrigation scheme and therefore would be counter to the goals of the study, which are to evaluate this particular scheme.

2) GVF is used to define the crop root zone, which impacts the amount of water applied by the irrigation scheme. The maximum root zone for each crop type is defined by a lookup table; the GVF is multiplied by the maximum root zone to determine the crop root zone. In this way, the scheme mimics the season cycle of crop root growth. More water is applied for greater crop root depth. Therefore, GVF is important for defining the irrigation season, triggering irrigation, and for determining the amount of irrigation water applied by the irrigation scheme.

The land surface model does not explicitly use a phenology dataset, such as MODIS EVI or NDVI, but rather uses proxies of Greenness Vegetation Fraction (GVF) and Leaf Area Index. The SPORT GVF dataset is based on NDVI, and therefore in essence translates the MODIS NDVI information into a form that the model can use (GVF).

11) Page 8 L12-15: You need to mention the resolution of these in- put datasets.

The resolution of each dataset has been added to this paragraph. It now reads as follows, with the additions shown in bold italics:

"Additional datasets common to all simulations include MODIS – International Geosphere Biosphere Program (MODIS-IGBP) land cover **at 1 km**, State Soil Geographic (STATSGO?) soil texture **at 1 km**, University of Maryland crop type **at 1 km**, and National Land Data Assimilation System – Phase 2 (NLDAS2, Xia et al., 2012) meteorological forcing at **1/8th degree (approximately 12 km)** that includes bias corrected radiation and gauge-based precipitation."

Is the UMD crop type product static or is a separate classification provided for each year? The annual Cropland Data Layer (https://www.nass.usda.gov/Research_and_Science/Cropland/SARS1a.php) product (provided at 30 m) is updated for each year to account for crop rotations and changing crop type patterns and might be a more correct source to use for something like this.

The UMD crop type product is static. We agree that the Cropland Data Layer is a great improvement on static crop maps and we have discussed integrating the CDL into LIS. However, for this study, because of the small domain and the detailed ground observations we have, the CDL would not have added value beyond the ground truth provided by the Franz group. We completed the default crop type run and an additional crop type run with an observationally tuned map (detailed in the Discussion section) and found no significant differences. As a result, we believe a run with the CDL would not have differed significantly from either of these two runs.

12) Page 9 L4-5: The GVF product is used for establishing the length and timing of the growing season. A more appropriate source for this would be the MODIS global vegetation phenology product (MCD12Q2) currently produced at 500 m resolution that is also more consistent with the LSM resolution and the CRNP validation dataset (and the scale of irrigation effects). Reasons for not using something like this should be addressed.

As discussed above, a main feature of the Ozdogan et al. (2010) irrigation scheme is the determination of the irrigation season based on a threshold of the GVF. While it is certainly possible to use a different method, such as the MODIS phenology for determining the irrigation season, this would be a considerable deviation from the scheme and therefore would be counter to the goals of the study in evaluating this particular scheme.

The land surface model does not explicitly use a phenology dataset, such as MODIS EVI or NDVI, but rather uses proxies of Greenness Vegetation Fraction (GVF) and Leaf Area Index. The SPORT GVF dataset is based on NDVI, and therefore essentially translates the MODIS NDVI information into a form that the LSM can use (GVF).

13) Page 9 Section 4: A brief intro statement would be useful here.

We don't believe an intro statement is necessary here as the previous paragraph sets up the organization of this section.

14) Page 10 L7: The relationship used to compute the root zone length from GVF should be provided in the methodology.

The root zone length calculation, as it applies to the irrigation scheme, is described on Page 10, Lines 7-8.

> "…,while the root zone is the produce of the maximum root depth (as defined by crop type) scaled by the GVF to mimic a seasonal cycle of root growth."

15) Page 12 L6: This is the first mentioning of a rainfed validation site within the study domain. Details like this should be provided in the method section (preferably in a dedicated study region section).

The rainfed site was mentioned in Section 2.3 but has been moved to the new Evaluation Data section (3.2).

16) Page 13 L8-13: This should be moved to the methodology section. A shorter summary of the CRNP would suffice here.

The description of the CRNP gridded soil moisture product and the alterations made to the regression for this study have been moved to the new Evaluation Data section (3.2)

17) Page 13 L15: Not clear what modifications were made to the COSMOS product; provide a section reference or more details here. Also a bit confused about the references to both CRNP and COSMOS as they are presumably the same thing?

COSMOS is the observing network, CRNP is the instrument. COSMOS was a typo here and has been corrected to 'CRNP'. More description has been added about the changes to the spatial regression and they've been moved to the new Evaluation Data section (3.2).

18) Page 13 L14-15: I wonder if a non-cumulative PDF wouldn't be better in this context?

This comment echoes that of reviewer 3 in that this information could be presented in a more effective manner. This figure has been changed to a scatterplot:

[Figure]

19) Page 14 L6: I believe that the GVF is provided at 3 km (and 16 km) rather than 1 km resolution, correct?

The SPoRT GVF is provided at 3 km, but the climatological GVF is provided at 1 km. Please see comment #10.

20) Section 5: The discussion is very brief and lacks more substantial and high quality discussion elements on limitations, challenges and opportunities.

A paragraph has been added to the Discussion that addresses the concerns of reviewers 2 and 3 related to the choice of meteorological forcing dataset. An additional paragraph has been added discussing the potential limitations of the uncoupled configuration used in this study.

Other limitations of the study are presented in the discussion Page 16, paragraph 2 and 3. Challenges are discussed extensively in the Background section. The future of irrigation intensity datasets is detailed in Page 17 paragraph 2.

21) Page 15 L3-8: These are useful details that should have been provided in the methodology or result sections

A description of the triggering datasets and exactly how they impact triggering is included in the methodology section (Page 10 Lines 1-17). The relative importance of the triggering datasets is included here, not in the methodology, because this is a main finding of the study.

22) Page 15 L9-12: Not sure I understand this correctly, particularly the part about the scaling by GVF being more important than changes in rooting depth.

The logic here is as follows. First, the maximum crop root zone is multiplied by the GVF (non-dimensional number 0-1) to mimic a seasonal cycle of root growth. The amount of water added by the irrigation scheme is then dependent on the depth of the crop root zone (more water applied for crops that have deeper roots). To determine the potential impact of crop rooting depth specification, we completed an additional run where we used an observationally tuned crop map and changed the maximum root depth of maize and soybeans. It was concluded that the impacts of the crop root depth on irrigation amounts and fluxes were insignificant compared to the influence of the scaling of the crop root zone.

23) Page 15 L13: The method for determining the start and end of the growing season hasn't been described anywhere, but it must be. Justifications for adopting that methodology (rather than relying on existing phenology products for instance) should also be provided.

The details of the irrigation season have been added to the method section when first introduced. Page 10, Lines 3-4 now reads:

> "The growing season, addressed in question three, is a function of the gridcell GVF (i.e., 40% annual range in climatological GVF; Ozdogan et al. 2010)…"

This method is used as it is a main feature of the Sprinkler irrigation algorithm. Please see comment 12.

**Technical corrections:**

1) Page 4 L1: "with a two different.." - should be "with two different.."
This has been changed.

2) Page 4 L23: "..water resources region. . ."?
This has been reworded to:

> "reproduce irrigation water usage within counties and water resource regions, respectively"

3) Page 5 L14: use "high resolution" rather than "high-resolution"

4) Figure 5: I would also show the irrigation amounts here as done in Figure 7. Why is the impact of irrigation high when no irrigation is applied (e.g., during rain events)?

The observed irrigation amounts are not shown because this figure is used to analyze only model results/datasets, not observations. It would be possible to show simulated irrigation amounts for all irrigation runs, but that would make the figure much more confusing/busy without contributing additional information. We feel that the combination of forcing precipitation and flux changes due to irrigation already make it readily apparent when irrigation is being triggered.

As compared to the rain-free periods, the impact of irrigation is dramatically reduced during rain events. There is still some impact to fluxes during rain events in the summer because the soil column in the irrigated simulation is generally wetter than control due to the memory of previous irrigation, even if irrigation does not occur on that day.

5) Figure 5: Issue with the legends – they are not consistent with what is shown; currently I can only distinguish two different line styles.

This figure shows changes from control in each model configuration for latent and sensible heat fluxes. Latent heat flux changes are shown in blue and sensible heat flux changes are shown in red. The line style corresponds to the model configuration. Therefore, the change from Control in latent heat flux when using irrigation and the SPoRT GVF dataset is shown in the blue dotted line. Only two lines are distinguishable because the Tuned and Standard configurations do not differ enough from each other at this scale to be distinguishable. This is a main conclusion shown in the figure.

6) Figure 5: a and b rather than top and bottom should be used for more precise figure referencing in the manuscript. This also applies to the other figures.

All figures have been updated to use the (a),(b), etc.